# TOWARDS LOSSLESS DATASET DISTILLATION VIA DIFFICULTY-ALIGNED TRAJECTORY MATCHING

**Ziyao Guo**[1,3,4] **Kai Wang**[1†] **George Cazenavette**[2] **Hui Li**[4] **Kaipeng Zhang**[3‡] **Yang You**[1‡]
[1]National University of Singapore    [2]Massachusetts Institute of Technology
[3]Shanghai Artificial Intelligence Laboratory    [4]Xidian University
gzyaftermath@outlook.com, {kai.wang, youy}@comp.nus.edu.sg, gcaz@mit.edu

## ABSTRACT

The ultimate goal of *Dataset Distillation* is to synthesize a small synthetic dataset such that a model trained on this synthetic set will perform **equally well** as a model trained on the full, real dataset. Until now, no method of Dataset Distillation has reached this completely lossless goal, in part because they only remain effective when the total number of synthetic samples is *extremely small*. Since only so much information can be contained in such a small number of samples, it seems that to achieve truly lossless dataset distillation, we must develop a distillation method that remains effective as the size of the synthetic dataset grows. In this work, we present such an algorithm and elucidate *why* existing methods fail to generate larger, high-quality synthetic sets. Current state-of-the-art methods rely on *trajectory-matching*, or optimizing the synthetic data to induce similar long-term training dynamics as the real data. We empirically find that the *training stage* of the trajectories we choose to match (*i.e.*, early or late) greatly affects the effectiveness of the distilled dataset. Specifically, early trajectories (where the teacher network learns *easy* patterns) work well for a low-cardinality synthetic set since there are fewer examples wherein to distribute the necessary information. Conversely, late trajectories (where the teacher network learns *hard* patterns) provide better signals for larger synthetic sets since there are now enough samples to represent the necessary complex patterns. Based on our findings, we propose to align the difficulty of the generated patterns with the size of the synthetic dataset. In doing so, we successfully scale trajectory matching-based methods to larger synthetic datasets, achieving lossless dataset distillation for the very first time. Code and distilled datasets are available at https://github.com/NUS-HPC-AI-Lab/DATM.

## 1 INTRODUCTION

Dataset distillation (DD) aims at distilling a large dataset into a small synthetic one, such that models trained on the distilled dataset will have similar performance as those trained on the original dataset. In recent years, several algorithms have been proposed for this important topic, such as gradient matching (Zhao et al., 2020; Kim et al., 2022; Zhang et al., 2023; Liu et al., 2023b), kernel inducing points (Nguyen et al., 2020; 2021), distribution matching (Wang et al., 2022; Zhao & Bilen, 2023; Zhao et al., 2023), and trajectory matching (Cazenavette et al., 2022; Cui et al., 2023; Du et al., 2023). So far, dataset distillation has achieved great success in the regime of extremely small synthetic sets. For example, MTT (Cazenavette et al., 2022) achieves 71.6% test accuracy on CIFAR-10 using only 1% of the original data size. This impressive performance led to its application in a variety of downstream tasks such as continual learning (Masarczyk & Tautkute, 2020; Rosasco et al., 2021), privacy protection (Zhou et al., 2020; Sucholutsky & Schonlau, 2021a; Dong et al., 2022; Chen et al., 2022; Xiong et al., 2023), and neural architecture search (Such et al., 2020; Wang et al., 2021).

However, although previous DD methods have achieved great success with very few IPC (images-per-class), there still remains a significant gap between the performance of their distilled datasets and the full, real counterparts. To minimize this gap, one would intuitively think to increase the size of the

---

[†]Project lead.
[‡]Corresponding author.

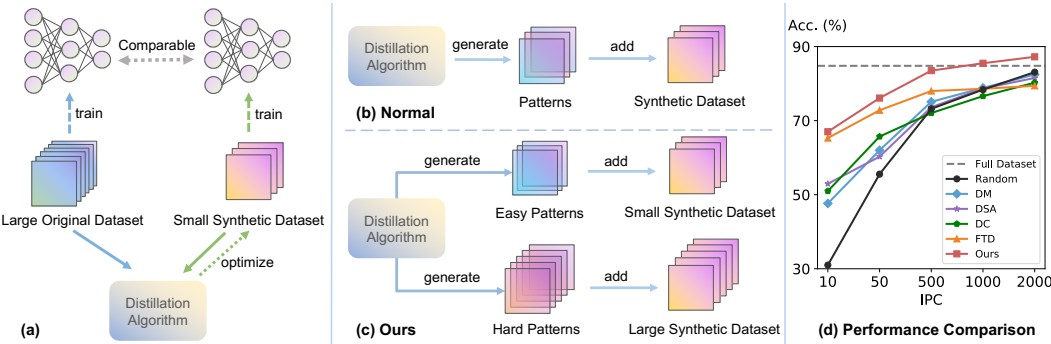

Figure 1: (a) Illustration of the objective of dataset distillation. (b) The optimization in dataset distillation can be viewed as the process of generating informative patterns on the synthetic dataset. (c) We align the difficulty of the synthetic patterns with the size of the distilled dataset, to enable our method to perform well in both small and large IPC regimes. (d) Comparison of the performance of multiple dataset distillation methods on CIFAR-10 with different IPC. As IPC increases, the performance of previous methods becomes worse than random selection.

synthetic dataset. Unfortunately, as IPC increases, previous distillation methods mysteriously become less effective, even performing worse than random selection (Cui et al., 2022; Zhou et al., 2023). In this paper, we offer an answer as to *why* previous dataset distillation methods become ineffective as IPC increases and, in doing so, become the first to circumvent this issue, allowing us to achieve lossless dataset distillation.

We start our work by observing the patterns learned by the synthetic data, taking trajectories matching (TM) based distillation methods (Cazenavette et al., 2022; Du et al., 2023) as an example. Generally, the process of dataset distillation can be viewed as the embedding of informative patterns into a set of synthetic samples. For TM-based distillation methods, the synthetic data learns patterns by matching the training trajectories of surrogate models optimized over the synthetic dataset and the real one. According to (Arpit et al., 2017), deep neural networks (DNNs) typically learn to recognize `easy` patterns early in training and `hard` patterns later on. As a result, we note that the properties of the data generated by TM-based methods vary widely depending on from which teacher training stage we sample our trajectories from (early or late). Specifically, matching early or late trajectories causes the synthetic data to learn `easy` or `hard` patterns respectively.

We then empirically show that the effect of learning `easy` and `hard` patterns varies with the size of the synthetic dataset (*i.e.*, IPC). In low-IPC settings, `easy` patterns prove the most beneficial since they explain a larger portion of the real data distribution than an equivalent number of `hard` samples. However, with a sufficiently large synthetic set, learning `hard` samples becomes optimal since their union covers both the `easy` and "long-tail" `hard` samples of the real data. In fact, learning `easy` patterns in the high-IPC setting performs *worse* than random selection since the synthetic images collapse towards the mean patterns of the distribution and can no longer capture the long-tail parts. Previous distillation methods default toward distilling `easy` patterns, leading to their ineffectiveness in high-IPC cases.

The above findings motivate us to manage to align the difficulty of the learned patterns with the size of the distilled dataset, in order to keep our method effective in both low and high IPC cases. Our experiments show that, for TM-based methods, we can control the difficulty of the generated patterns by only matching the trajectories of a specified training phase. By doing so, our method is able to work well in both low and high IPC settings. Furthermore, we propose to learn `easy` and `hard` patterns sequentially, making the optimization stable enough for learning soft labels during the distillation, bringing further significant improvement. Our method achieves state-of-the-art performance in both low and high IPC cases. Notably, we distill CIFAR-10 and CIFAR-100 to 1/5 and Tiny ImageNet to 1/10 of their original sizes without any performance loss on ConvNet, offering the first lossless method of dataset distillation.

## 2 PRELIMINARY

For a given large, real dataset $\mathcal{D}_{\text{real}}$, dataset distillation aims to synthesize a smaller dataset $\mathcal{D}_{\text{syn}}$ such that models trained on $\mathcal{D}_{\text{syn}}$ will have similar test performance as models trained on $\mathcal{D}_{\text{real}}$.

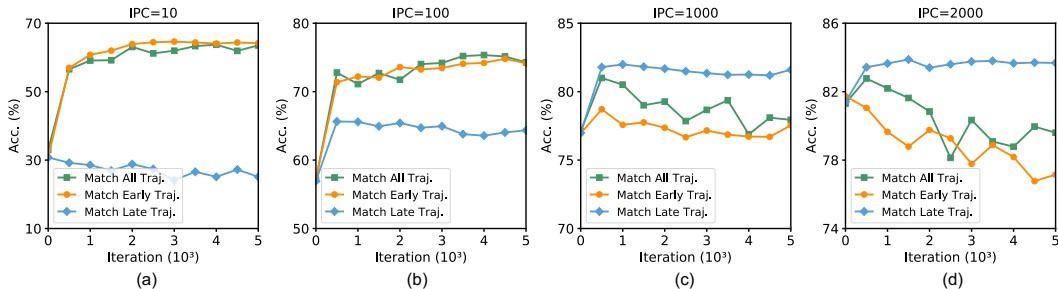

Figure 2: We train expert models on CIFAR-10 for 40 epochs. Then the distillation is performed under different IPC settings by matching either early trajectories $\{\theta_t^*|0 \leq t \leq 20\}$, late trajectories $\{\theta_t^*|20 \leq t \leq 40\}$, or all trajectories $\{\theta_t^*|0 \leq t \leq 40\}$. As IPC increases, matching late trajectories becomes beneficial while matching early trajectories tends to be harmful.

For trajectory matching (TM) based methods, the distillation is performed by matching the training trajectories of the surrogate models optimized over $\mathcal{D}_{\text{real}}$ and $\mathcal{D}_{\text{syn}}$. Specifically, let $\tau^*$ denote the expert training trajectories, which is the time sequence of parameters $\{\theta_t^*\}_0^n$ obtained during the training of a network on the real dataset $\mathcal{D}_{\text{real}}$. Similarly, $\hat{\theta}_t$ denotes the parameters of the network trained on the synthetic dataset $\mathcal{D}_{\text{syn}}$ at training step $t$.

In each iteration of the distillation, $\theta_t^*$ and $\theta_{t+M}^*$ are randomly sampled from a set of expert trajectories $\{\tau^*\}$ as the start parameters and target parameters used for the matching, where $M$ is a preset hyper-parameter. Then TM-based distillation methods optimize the synthetic dataset $\mathcal{D}_{\text{syn}}$ by minimizing the following loss:

$$\mathcal{L} = \frac{\|\hat{\theta}_{t+N} - \theta_{t+M}^*\|_2^2}{\|\theta_t^* - \theta_{t+M}^*\|_2^2},\tag{1}$$

where $N$ is a preset hyper-parameter and $\hat{\theta}_{t+N}$ is obtained in the inner optimization with cross-entropy (CE) loss $\ell$ and the trainable learning rate $\alpha$:

$$\hat{\theta}_{t+i+1} = \hat{\theta}_{t+i} - \alpha \nabla \ell(\hat{\theta}_{t+i}, \mathcal{D}_{\text{syn}}), \text{where } \hat{\theta}_t := \theta_t^*.\tag{2}$$

## 3 METHOD

In this section, we first analyze the influence of matching trajectories from different training stages. Then, we introduce our method and its carefully designed modules.

### 3.1 EXPLORATION

TM-based methods generate patterns on the synthetic data by matching training trajectories. According to Arpit et al. (2017), DNNs tend to learn `easy` patterns early in training, then the harder ones later on. Motivated by this, we start our work by exploring the effect of matching trajectories from different training phases. Specifically, we train expert models for 40 epochs and roughly divide their training trajectories into two parts: the early trajectories $\{\theta_t^*|0 \leq t \leq 20\}$ and the latter ones $\{\theta_t^*|20 \leq t \leq 40\}$. Then we perform the distillation by matching these two sets of trajectories under various IPC settings. Experimental results are reported in Figure 2. Our observations and relevant analyses are presented as follows.

**Observation 1.** As shown in Figure 2, matching early trajectories works better with small synthetic datasets, but matching late trajectories performs better as the size of the synthetic set grows larger.

**Analysis 1.** Since DNNs learn to recognize `easy` patterns early in training and `hard` patterns later on, we infer that matching early trajectories yields distilled data with `easy` patterns while matching late trajectories produces hard ones. Combined with the empirical results from Figure 2, we can conclude that distilled data with `easy` patterns perform well for small synthetic sets while data with `hard` features work better with larger sets. Perhaps unsurprisingly, this highly coincides with a common observation in the area of *dataset pruning*: preserving easy samples works better when very

few samples are kept, while keeping hard samples works better when the pruned dataset is larger (Sorscher et al., 2022).

**Observation 2.** As can be observed in Figure 2 (a), matching late trajectories leads to poor performance in the low IPC setting. When IPC is high, matching early trajectories will consistently undermine the performance of the synthetic dataset as the distillation goes on, as can be observed in Figure 2 (d). Also, as reflected in Figure 2, simply choosing to match all trajectories is not a good strategy.

**Analysis 2.** In low IPC settings, due to distilled data's limited capacity, it is challenging to learn data that models the outliers (`hard` samples) without neglecting the more plentiful `easy` samples; since `easy` samples make up most of the real data distribution, modeling these samples is more efficient performance-wise when IPC is low. Therefore, matching early trajectories (which will generate `easy` patterns) performs better than matching later ones (for low IPC). Conversely, in high IPC settings, distilling data that models only the `easy` samples is no longer necessary, and will even perform worse than a random subset of real samples. Thus, we must now consider the less-common `hard` samples by matching late trajectories (Figure 4). Since previous distillation methods focus on extremely small IPC cases, they tend to be biased towards generating `easy` patterns, leading to their ineffective in large IPC cases.

Based on the above analyses, to keep dataset distillation effective in both low and high IPC cases, we must calibrate the difficulty of the generated patterns (*i.e.*, avoid generating patterns that are too easy or too difficult). To this end, we propose our method: Difficulty-Aligned Trajectory Matching, or DATM.

## 3.2 DIFFICULTY-ALIGNED TRAJECTORY MATCHING

Since patterns learned by matching earlier trajectories are easier than the later ones, we can control the difficulty of the generated patterns by restricting the trajectory-matching range. Specifically, let $\tau^* = \{\theta_t^* | 0 \leq t \leq n\}$ denote an expert trajectory. To control the matching range flexibly, we set a lower bound $T^-$ and an upper bound $T^+$ on the sample range of $t$, such that only parameters within $\{\theta_t^* | T^- \leq t \leq T^+\}$ can be sampled for the matching. Then the trajectory segment used for the matching can be formulated as:

$$\tau^* = \{\underbrace{\theta_0^*, \theta_1^*, \cdots,}_{\text{too easy}} \underbrace{\theta_{T^-}^*, \cdots, \theta_{T^+}^*,}_{\text{matching range}} \underbrace{\cdots, \theta_n^*}_{\text{too hard}}\}. \tag{3}$$

To further enrich the information contained in the synthetic dataset, an intuitive choice is using soft labels (Hinton et al., 2015). Recently, Cui et al. (2023) show that using soft labels to guide the distillation can bring non-trivial improvement for the performance. However, their soft labels are not optimized during the distillation, leading to poor consistency between synthetic data and soft labels. To enable learning labels, we find the following challenges need to be solved:

**Mislabeling.** We use logits $L_i = f_{\theta^*}(x_i)$ to initialize soft labels, which are generated by the pre-trained model $f_{\theta^*}$ sampled from expert trajectories. However, labels initialized in this way might be incorrect (*i.e.*, target class doesn't have the highest logit score). To avoid mislabeling, we sift through $\mathcal{D}_{\text{real}}$ to find samples that can be correctly classified by model $f_{\theta^*}$ and use them to construct the subset $\mathcal{D}_{\text{sub}}$. Then we randomly select samples from $\mathcal{D}_{\text{sub}}$ to initialize $\mathcal{D}_{\text{syn}} = \{(x_i, \hat{y}_i = \text{softmax}(L_i))\}$, such that we can avoid the distillation being misguided by the wrong label.

**Instability.** During the experiments, we found that optimizing soft labels will increase the instability of the distillation when the IPC is low. In low IPC settings, the distillation loss tends to be higher and less stable overall since the smaller synthetic set struggles to induce a proper training trajectory. This issue becomes fatal when labels are optimized during the distillation, as the labels are too fragile to take the wrong guidance brought by the mismatch, leading to increased instability. To alleviate this, we propose to generate only `easy` patterns in the early distillation phase. After enough `easy` patterns are embedded into the synthetic data for surrogate models to learn them well, we then gradually generate harder ones. By applying this sequential generation (SG) strategy, the surrogate model can match the expert trajectories better. Accordingly, the distillation becomes more stable.

In practice, to generate only `easy` patterns at the early distillation stage, we set a floating upper bound $T$ on the sample range of $t$, which is set to be relatively small in the beginning and will be gradually increased as the distillation progresses until it reaches its upper bound $T^+$. Overall, the

| Dataset | CIFAR-10 | | | | | CIFAR-100 | | | | Tiny ImageNet | | |
|---|---|---|---|---|---|---|---|---|---|---|---|---|
| IPC | 1 | 10 | 50 | 500 | 1000 | 1 | 10 | 50 | 100 | 1 | 10 | 50 |
| Ratio | 0.02 | 0.2 | 1 | 10 | 20 | 0.2 | 2 | 10 | 20 | 0.2 | 2 | 10 |
| Random | 15.4±0.3 | 31.0±0.5 | 50.6±0.3 | 73.2±0.3 | 78.4±0.2 | 4.2±0.3 | 14.6±0.5 | 33.4±0.4 | 42.8±0.3 | 1.4±0.1 | 5.0±0.2 | 15.0±0.4 |
| DC | 28.3±0.5 | 44.9±0.5 | 53.9±0.5 | 72.1±0.4 | 76.6±0.3 | 12.8±0.3 | 25.2±0.3 | - | - | - | - | - |
| DM | 26.0±0.8 | 48.9±0.6 | 63.0±0.4 | 75.1±0.3 | 78.8±0.1 | 11.4±0.3 | 29.7±0.3 | 43.6±0.4 | - | 3.9±0.2 | 12.9±0.4 | 24.1±0.3 |
| DSA | 28.8±0.7 | 52.1±0.5 | 60.6±0.5 | 73.6±0.3 | 78.7±0.3 | 13.9±0.3 | 32.3±0.3 | 42.8±0.4 | - | - | - | - |
| CAFE | 30.3±1.1 | 46.3±0.6 | 55.5±0.6 | - | - | 12.9±0.3 | 27.8±0.3 | 37.9±0.3 | - | - | - | - |
| KIP[1] | 49.9±0.2 | 62.7±0.3 | 68.6±0.2 | - | - | 15.7±0.2 | 28.3±0.1 | - | - | - | - | - |
| FRePo[1] | 46.8±0.7 | 65.5±0.4 | 71.7±0.2 | - | - | 28.7±0.1 | 42.5±0.2 | 44.3±0.2 | - | 15.4±0.3 | 25.4±0.2 | - |
| RCIG[1] | 53.9±1.0 | 69.1±0.4 | 73.5±0.3 | - | - | 39.3±0.4 | 44.1±0.4 | 46.7±0.3 | - | 25.6±0.3 | 29.4±0.2 | - |
| MTT[2] | 46.2±0.8 | 65.4±0.7 | 71.6±0.2 | ⬉ | ⬉ | 24.3±0.3 | 39.7±0.4 | 47.7±0.2 | 49.2±0.4 | 8.8±0.3 | 23.2±0.2 | 28.0±0.3 |
| TESLA[2] | **48.5±0.8** | 66.4±0.8 | 72.6±0.7 | ⬉ | ⬉ | 24.8±0.4 | 41.7±0.3 | 47.9±0.3 | 49.2±0.4 | - | - | - |
| FTD[2,3] | 46.0±0.4 | 65.3±0.4 | 73.2±0.2 | ⬉ | ⬉ | 24.4±0.4 | 42.5±0.2 | 48.5±0.3 | 49.7±0.4 | 10.5±0.2 | 23.4±0.3 | 28.2±0.4 |
| **DATM** (Ours) | 46.9±0.5 | **66.8±0.2** | **76.1±0.3** | **83.5±0.2** | **85.5±0.4** | **27.9±0.2** | **47.2±0.4** | **55.0±0.2** | **57.5±0.2** | **17.1±0.3** | **31.1±0.3** | **39.7±0.3** |
| Full Dataset | | 84.8±0.1 | | | | | 56.2±0.3 | | | | 37.6±0.4 | |

Table 1: Comparison with previous dataset distillation methods on CIFAR-10, CIFAR-100 and Tiny ImageNet. ConvNet is used for the distillation and evaluation. Hilighted results indicate we achieve lossless distillation. Our method consistently out-performs prior works and is the only to achieve lossless distillation.
[1]Kernel-based methods use a much larger neural network; we underline their results when they perform best.
[2]Previous TM-based methods perform *worse* than random initialization in higher IPC cases, indicated by ⬉.
[3]For a fair comparison, we reproduce FTD without using EMA (exponential moving average).

process of sampling the start parameters $\theta_t^*$ can be formulated as:

$$\theta_t^* \sim \mathcal{U}(\{\theta_{T^-}^*, \cdots, \theta_T^*\}), \text{ where } T \to T^+. \tag{4}$$

In each iteration, after deciding the value of $t$, we then sample $\theta_t^*$ and $\theta_{t+M}^*$ from expert trajectories as the start parameters and the target parameters for the matching. Then $\hat{\theta}_{t+N}$ can be obtained by Eq. 2. Subsequently, after calculating the matching loss using Eq. 1, we perform backpropagation to calculate the gradients and then use them to update the synthetic data $x_i$ and $L_i$, where $(x_i, \hat{y}_i = \text{softmax}(L_i)) \in \mathcal{D}_{\text{syn}}$. See Algorithm 1 for the pseudocode of our method.

# 4 EXPERIMENTS

## 4.1 SETUP

We compare our method with several representative distillation methods including DC (Zhao et al., 2020), DM (Zhao & Bilen, 2023), DSA (Zhao & Bilen, 2021), CAFE (Wang et al., 2022), KIP (Nguyen et al., 2020), FRePo (Zhou et al., 2022), RCIG (Loo et al., 2023), MTT (Cazenavette et al., 2022), TESLA (Cui et al., 2023), and FTD (Du et al., 2023). The evaluations are performed on several popular datasets including CIFAR-10, CIFAR-100 (Krizhevsky et al., 2009), and Tiny ImageNet (Le & Yang, 2015). We generate expert trajectories in the same way as FTD without modifying the involved hyperparameters. We also use the same suite of differentiable augmentations (Zhao & Bilen, 2021) in the distillation and evaluation stage, which is generally utilized in previous works (Zhao & Bilen, 2021; Wang et al., 2022; Cazenavette et al., 2022; Du et al., 2023).

Consistent with previous works, we use networks with instance normalization by default, while networks with batch normalization are indicated with "-BN" (*e.g.*, ConvNet-BN). Without particular specification, we perform distillation using a 3-layer ConvNet for CIFAR-10 and CIFAR-100, while we move up to a depth-4 ConvNet for Tiny ImageNet. We also use LeNet (LeCun et al., 1998), AlexNet (Krizhevsky et al., 2012), VGG11 (Simonyan & Zisserman, 2015), and ResNet18 (He et al., 2016) for cross-architecture experiments. More details can be found in Section A.8.

## 4.2 MAIN RESULTS

**CIFAR-10/100 and Tiny ImageNet**. As the results reported in Table 1, our method outperforms other methods with the same network architecture in all settings but CIFAR-10 with IPC=1. As can be observed, the improvements brought by previous distillation methods are quickly saturated as the distillation ratio approaches 20%. Especially in CIFAR-10, almost all previous methods have similar or even worse performance than random selection when the ratio is greater than 10%. Benefiting from our difficulty alignment strategy, our method remains effective in high IPC cases. Notably, we

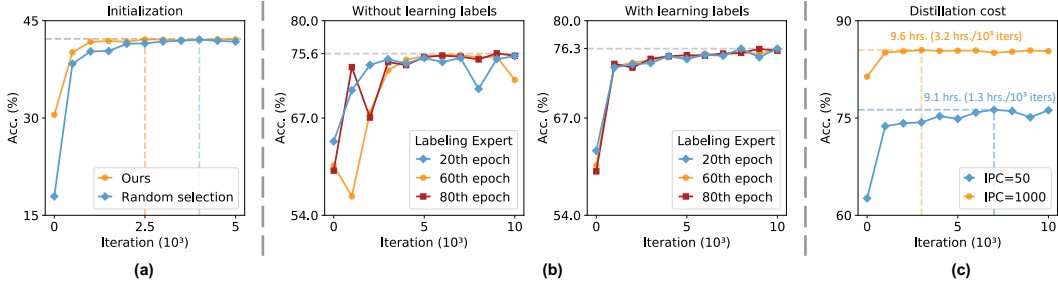

Figure 3: **(a)**: (CIFAR-100, IPC=10) Synthetic datasets are initialized by randomly sampling data from the original dataset (random selection) or a subset of data that can be correctly classified (ours). Our strategy makes the optimization converge faster. **(b)**: (CIFAR-10, IPC=50) Ablation on learning soft labels, where soft labels are initialized with expert models trained after different epochs. Learning labels relieves us from carefully selecting the labeling expert. **(c)**: (CIFAR-10) The optimization with higher IPC converges in fewer iterations.

| Method | ConvNet | ResNet18 | VGG | AlexNet |
|---|---|---|---|---|
| Random | 33.46 | 31.95 | 32.18 | 26.65 |
| MTT | 45.68 | 42.56 | 41.22 | 40.29 |
| FTD | 48.90 | 46.65 | 43.24 | 42.20 |
| **DATM** | **55.03** | **51.71** | **45.38** | **45.74** |

(a) CIFAR-100, IPC=50

| Soft Label | Difficulty Alignment | Acc |
|---|---|---|
|  |  | 48.50 |
|  | ✓ | 50.79 |
| ✓ |  | 52.96 |
| ✓ | ✓ | 55.03 |

(b) CIFAR-100, IPC=50

| Label Learning | Sequential Gen. | Acc |
|---|---|---|
|  |  | 72.8 |
| ✓ |  | 75.0 |
|  | ✓ | 75.6 |
| ✓ | ✓ | 76.1 |

(c) CIFAR-10, IPC=50

Table 2: **(a)**: Cross-Architecture evaluation. Our distilled dataset performs well across various unseen networks. **(b)**: Ablation studies on the components of our method; all bring non-trivial improvement. **(c)**: Ablation on learning soft labels and our sequential generation (SG) strategy.

successfully distill CIFAR-10 and CIFAR-100 to 1/5, and Tiny ImageNet to 1/10 their original size without causing any performance drop.

**Cross-architecture generalization.** Here we evaluate the generalizability of our distilled datasets in various IPC settings. As the results reported in Table 2 (Left), our distilled dataset performs best on unseen networks when IPC is small, reflecting the good generalizability of the data and labels distilled by our method. Furthermore, we evaluate the generalizability in high IPC settings and compare the performance with two representative coreset selection methods including Glister (Killamsetty et al., 2021) and Forgetting (Toneva et al., 2018). As shown in Table 4, although coreset selection methods are applied case by case, they are not universally beneficial for all networks. Notably, although our synthetic dataset is distilled with ConvNet, it generalizes well on all networks, bringing non-trivial improvement. Furthermore, on CIFAR-100, the improvement of AlexNet is even higher than that of ConvNet. This reflects the overfitting problem of synthetic datasets to distillation networks is somewhat alleviated in higher IPC situations.

## 4.3 ABLATION

**Ablation on components of our method.** We perform ablation studies by adding the components of our methods one by one to measure their effect. As the results reported in Table 2 (b,c), all the components of our method bring non-trivial improvement. Especially, when soft labels are utilized, the distillation becomes unstable if our proposed sequential generation strategy is not applied, leading to poor performance and sometimes program crashes.

**Soft label.** In our method, we use logits generated by the pre-trained model to initialize soft labels, since having an appropriate distribution before the softmax is critical for the optimization of the soft label (Section A.2). However, using logits will introduce additional information to the distilled dataset (Hinton et al.,

| Dataset | CIFAR-10 | | | CIFAR-100 | | | Tiny ImageNet | | |
|---|---|---|---|---|---|---|---|---|---|
| IPC | 1 | 10 | 50 | 1 | 10 | 50 | 1 | 10 | 50 |
| FTD | 46.0 | 65.3 | 73.2 | 24.4 | 42.5 | 48.5 | 10.5 | 23.4 | 28.2 |
| FTD+ASL | 45.5 | 66.1 | 72.8 | 24.2 | 44.5 | 51.2 | 12.4 | 26.7 | 30.9 |
| **DATM** | **46.9** | **66.8** | **76.1** | **27.9** | **47.2** | **53.0** | **17.1** | **29.0** | **33.7** |

Table 3: We assign soft labels (ASL) for datasets distilled by FTD. For fairness, our difficult alignment strategy is not utilized here. Results in red indicate the case when ASL is harmful.

| Dataset | Ratio | Method | ConvNet | ConvNet-BN | ResNet18 | ResNet18-BN | VGG11 | AlexNet | LeNet | MLP | Avg. |
|---|---|---|---|---|---|---|---|---|---|---|---|
| CIFAR-10 | 20% | Random | 78.38 | 80.25 | 84.58 | 87.21 | 80.81 | 80.75 | 61.85 | 50.98 | 75.60 |
| | | Glister | 62.46 | 70.52 | 81.10 | 74.59 | 78.07 | 70.55 | 56.56 | 40.59 | 66.81 |
| | | Forgetting | 76.27 | 80.06 | 85.67 | 87.18 | 82.04 | 81.35 | 64.59 | 52.21 | 76.17 |
| | | **DATM** | **85.50** | **85.23** | **87.22** | **88.13** | **84.65** | **85.14** | **66.70** | **52.40** | **79.37** |
| | | ↑ | +7.12 | +4.98 | +2.64 | +0.92 | +3.84 | +4.39 | +4.85 | +1.42 | +3.77 |
| CIFAR-100 | 20% | Random | 42.80 | 46.38 | 47.48 | 55.62 | 42.69 | 38.05 | 25.91 | 20.66 | 39.95 |
| | | Glister | 35.45 | 37.13 | 42.49 | 46.14 | 43.06 | 28.58 | 23.33 | 17.08 | 34.16 |
| | | Forgetting | 45.52 | 49.99 | 51.44 | 54.65 | 43.28 | 43.47 | 27.22 | 22.90 | 42.30 |
| | | **DATM** | **57.50** | **57.75** | **57.98** | **63.34** | **55.10** | **55.69** | **33.57** | **26.39** | **50.92** |
| | | ↑ | +14.70 | +11.37 | +10.50 | +7.72 | +12.41 | +17.64 | +7.66 | +5.73 | +10.97 |
| TI | 10% | Random | 15.00 | 24.21 | 17.73 | 28.07 | 22.51 | 14.03 | 9.25 | 5.85 | 17.08 |
| | | Glister | 17.32 | 19.77 | 18.84 | 23.12 | 19.10 | 11.68 | 8.84 | 3.86 | 15.32 |
| | | Forgetting | 20.04 | 23.83 | 19.38 | 28.88 | 23.77 | 12.13 | 12.06 | 5.54 | 18.20 |
| | | **DATM** | **39.68** | **40.32** | **36.12** | **43.14** | **38.35** | **35.10** | **12.41** | **9.02** | **31.76** |
| | | ↑ | +24.68 | +16.11 | +18.39 | +15.07 | +15.84 | +21.07 | +3.16 | +3.17 | +14.68 |

Table 4: We evaluate our distilled lossless datasets on unseen networks and compare them with two coreset selection methods. Results worse than random selection are indicated with red color. ↑ denotes the performance improvement brought by our method compared with random selection. TI denotes Tiny ImageNet.

2015). To see if this information can be directly integrated into the distilled datasets, we assign soft labels for datasets distilled by FTD (Du et al., 2023).

As shown in Table 3, directly assigning soft labels for the distilled datasets will even hurt its performance when the number of categories in the classification problem is small. For CIFAR-100 and Tiny ImageNet, although assigning soft labels slightly improves the performance of FTD, there is still a huge gap between its performance and ours. This is because the soft labels synthesized by our method are optimized constantly during the distillation, leading to better consistency between the synthetic data and their labels.

Furthermore, the information contained in logits varies with the capacity of the teacher model (Zong et al., 2023; Cui et al., 2023). In the experiments reported in Figure 3 (b), we use models trained after different epochs to initialize the soft labels. In the beginning, this difference has non-trivial influences on the performance of the synthetic datasets. However, the performance gaps soon disappear as the distillation goes on if labels are optimized during the distillation. This indicates learning labels relieves us from carefully selecting models to initialize soft labels. Moreover, as can be observed in Figure 3 (b), when soft labels are not optimized, the distillation becomes less stable, leading to the poor performance of the distilled dataset. Because using unoptimized soft labels will enlarge the discrepancy between the training trajectories over the synthetic dataset and the original one, considering the experts are trained with one-hot labels. More analyses are attached in Section A.2.2.

**Synthetic data initialization.** To avoid mislabeling, we initialize the synthetic dataset by randomly sampling data from a subset of the original dataset, which only contains samples that can be correctly classified by the model used for initializing soft labels. The process of constructing the subset can be viewed as a simple coreset selection. Here we perform an ablation study to see its effect. Specifically, synthetic datasets are either initialized by randomly sampling data from the original dataset (random selection) or a subset of data that can be correctly classified by a pre-trained ConvNet (ours).

As shown in Fig 3 (a), our initialization strategy can significantly speed up the convergence of the optimization. This is because data selected by our strategy are relatively easier for DNNs to learn. Thus, models trained on these easier samples will perform better when only limited training data are provided (Sorscher et al., 2022). Although this gap is gradually bridged as the distillation goes on, our initialization strategy can be utilized as a distillation speed-up technique.

## 5 EXTENSION

### 5.1 VISUALIZATION

For a better understanding of `easy` patterns and `hard` patterns, we visualize the distilled images and discuss their properties. In Figure 4, we visualize the images synthesized by matching early trajectories and late trajectories under the same IPC setting, where `easy` patterns and `hard` ones are learned respectively. In Figure 5, we visualize the images distilled under different IPC settings.

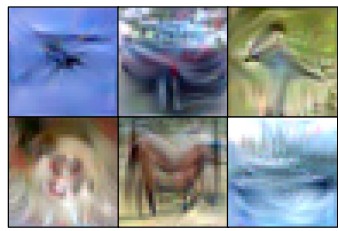 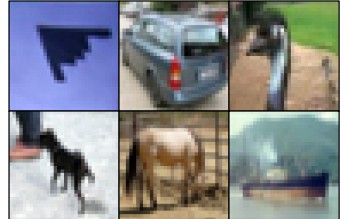 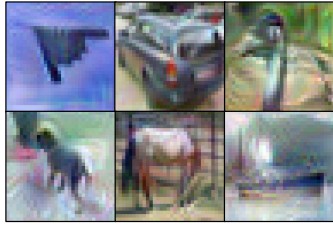

Match Early Trajectories          Original Images          Match Late Trajectories

Figure 4: We perform the distillation on CIFAR-10 with IPC=50 by matching either early trajectories $\{\theta_t | 0 \leq t \leq 10\}$ or late trajectories $\{\theta_t | 30 \leq t \leq 40\}$. All synthetic images are optimized 1000 times. Matching earlier trajectories will blur the details of the target object and change the color more drastically.

| E\D | Random | ConvNet | ResNet18 |
|---|---|---|---|
| ConvNet | 31.00 | **68.28** | 44.54 |
| ResNet18 | 26.70 | **48.66** | 45.28 |
| VGG11 | 31.63 | **45.93** | 39.68 |
| MLP | 26.86 | **33.39** | 29.17 |

IPC=10

| E\D | Random | ConvNet | ResNet18 |
|---|---|---|---|
| ConvNet | 55.55 | **76.08** | 59.08 |
| ResNet18 | 54.96 | **66.27** | 61.18 |
| VGG11 | 48.72 | **59.43** | 50.72 |
| MLP | **36.66** | 33.29 | 34.41 |

IPC=50

| E\D | Random | ConvNet | ResNet18 |
|---|---|---|---|
| ConvNet | 78.38 | **85.50** | 84.64 |
| ResNet18 | 84.58 | 87.22 | **87.70** |
| VGG11 | 80.81 | 84.65 | **84.85** |
| MLP | 50.98 | 52.40 | **54.01** |

IPC=1000

Table 5: We use ConvNet and ResNet18 to perform the distillation (D) on CIFAR-10 with various IPC settings. Then evaluations (E) are performed using networks with various architectures. As IPC increases, datasets distilled using ResNet18 perform relatively better.

As can be observed in Figure 4, compared with `hard` patterns, the learned `easy` patterns drive the synthetic images to move farther from their initialization and tend to blend the target object into the background. Although this process seems to make the image more informative, it blurs the texture and fine geometric details of the target object, making it harder for networks to learn to identify non-typical samples. This helps explain why generating `easy` patterns turned out to be harmful in high IPC cases. However, generating `easy` patterns performs well in the regime of low IPC, where the optimal solution is to model the most dense areas of the target category's distribution given the limited data budget. For example, as shown in Figure 5, the synthetic images collapse to almost only contain color and vague shape information when IPC is extremely low, which helps networks learn to identify `easy` samples from this basic property.

Furthermore, we find matching late trajectories yields distilled images that contain more fine details. For example, as can be observed in Figure 4, matching late trajectories transforms the simple background in the *dog* images into a more informative one and gives more texture details to the *dog* and *horse*. This transformation helps networks learn to identify outlier (`hard`) samples; hence, matching late trajectories is a better choice in high IPC cases.

## 5.2 DISTILLATION COST

In this work, we scale dataset distillation to high IPC cases. Surprisingly, we find the distillation cost does not increase linearly with IPC, since the optimization converges faster in large IPC cases. This is because we match only late trajectories in high IPC cases, where the learned `hard` patterns only make a few changes on the images, as we have analyzed in Section 5.1 and can be observed in Figure 5. In practice, as reflected in Figure 3 (c), although the distillation with IPC=1000 needs to optimize 20x more data than the case with IPC=50, the former one's cost is only 1.05 times higher.

## 5.3 DISTILLATION BACKBONE NETWORKS

So far, almost all representative distillation methods choose to use ConvNet to perform the distillation (Zhao & Bilen, 2021; Cazenavette et al., 2022; Loo et al., 2023). Using other networks as the distillation backbone will result in non-trivial performance degradation (Liu et al., 2023b). What makes ConvNet so effective for distillation remains an open question.

Here we offer an answer from our perspective: part of the specialness of ConvNet comes from its low capacity. In general, networks with more capacity can learn more complex patterns. Accordingly, when used as distilling networks, their generated patterns are relatively harder for DNNs to learn.

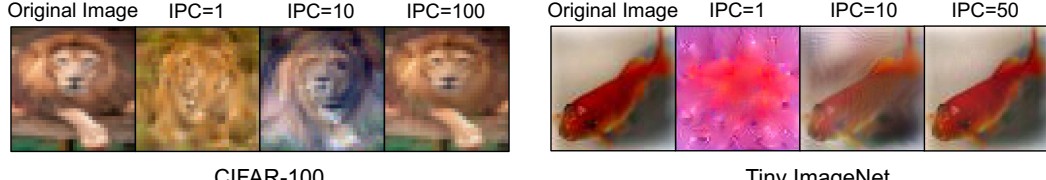

Figure 5: Visualization of the synthetic datasets distilled with different IPC settings. As IPC increases, synthetic images move less far from their initialization.

As we have analyzed in section 3.1, in small IPC cases (where previous distillation methods focus their attention), most improvement comes from the `easy` patterns generated on the synthetic data. Thus networks with more capacity such as ResNet will perform worse than ConvNet when IPC is low because their generated patterns are `harder` for DNNs to learn. However, in high IPC cases, where `hard` patterns play an important role, using stronger networks should perform relatively better. To verify this, we use ResNet18 and ConvNet to perform distillation on CIFAR-10 with different IPC settings. As shown in Table 5, when IPC is low, ConvNet performs much better than ResNet18 as the distillation network. However, when IPC reaches 1000, ResNet18 has a comparable or even better performance compared with ConvNet.

## 6 RELATED WORK

Dataset distillation introduced by Wang et al. (2018) is naturally a bi-level optimization problem, which aims at distilling a large dataset into a small one without causing performance drops. The following works can be divided into two types according to their mechanism:

**Kernel-based distillation methods** use kernel ridge-regression with NTK (Lee et al., 2019) to obtain a closed-form solution for the inner optimization (Nguyen et al., 2020). By doing so, dataset distillation can be formulated as a single-level optimization problem. The following works have significantly reduced the training cost (Zhou et al., 2022) and improved the performance (Loo et al., 2022; 2023). However, since the heavy resource consumption of inversing matrix operation, it is hard to scale kernel-based methods to larger IPC.

**Matching-based methods** minimize defined metrics of surrogate models learned from the synthetic dataset and the original one. According to the definition of the metric, they can be divided into four categories: based on matching gradients (Zhao et al., 2020; Kim et al., 2022; Zhang et al., 2023), features (Wang et al., 2022), distribution (Zhao & Bilen, 2023; Zhao et al., 2023), and training trajectories (Cazenavette et al., 2022; Cui et al., 2023; Du et al., 2023). So far, trajectory matching-based methods have shown impressive performance on every benchmark with low IPC (Cui et al., 2022; Yu et al., 2023). In this work, we further explore and show its great power in higher IPC cases.

## 7 CONCLUSION AND DISCUSSION

In this work, we find the difficulty of patterns generated by dataset distillation algorithms should be aligned with the size of the synthetic dataset, which is the key to keeping them effective in both low- and high-IPC cases. Building upon this insight, our method excels not only in low IPC cases but also maintains its efficacy in high IPC scenarios, achieving lossless dataset distillation for the first time.

However, our distilled data are only *lossless* for the distillation backbone network: when evaluating them with other networks, the performance drops still exist. We think this is because models with different capacities need varying amounts of training data. How to overcome this issue is still a challenging problem. Moreover, it is hard to scale TM-based methods to large datasets due to its high distillation cost. How to improve its efficiency would be the goal of our future work.

**Acknowledgements.** This work is supported by the National Research Foundation, Singapore under its AI Singapore Programme (AISG Award No: AISG2-PhD-2021-08- 008). This work is also supported in part by the National Key R&D Program of China (NO.2022ZD0160100 and NO.2022ZD0160101). Part of Hui Li's work is supported by the National Natural Science Foundation of China (61932015), Shaanxi Innovation Team project (2018TD-007), Higher Education Discipline Innovation 111 project (B16037). Yang You's research group is being sponsored by NUS startup grant (Presidential Young Professorship), Singapore MOE Tier-1 grant, ByteDance grant, ARCTIC grant, SMI grant (WBS number: A-8001104-00-00), Alibaba grant, and Google grant for TPU usage. We thank Bo Zhao for valuable discussions and feedback.

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

# A APPENDIX

## A.1 SOFT LABEL DISTRIBUTION

To observe the changes in soft labels' distribution during the distillation, we record the standard deviation (std) of soft labels (after softmax) for each synthetic image, and report the average value of their std in Figure 6. As can be observed, for all datasets, their labels' std tends to increase as the distillation goes on. However, this increase does not arise due to the diversity between the values of soft labels becoming larger, but the values of non-target categories' labels are suppressed. Since the value of the target category is much higher than others, to facilitate observation, we only report the values of non-target categories in Figure 7. As can be observed, after the distillation, the values of non-target categories' labels are suppressed more drastically when IPC is smaller. This is because, in low IPC cases, basic patterns of target category are embedded into the synthetic data since only early expert trajectories are used for the matching. Accordingly, the model becomes more confident that the generated sample belongs to the target category.

## A.2 SOFT LABEL INITIALIZATION

We have tried to initialize soft labels with the original one-hot labels and directly optimize their values during the distillation, but the distillation soon crashed. We also have tried to add a softmax layer on it. However, the distillation is still not stable. After initializing labels with class probabilities calculated using softmax and logits output by a pre-trained model, finally, soft labels can be optimized stably during the distillation.

For labels with a given distribution, their values before softmax can be different. In this case, due to the utilization of softmax, when performing backpropagation, the gradients of pre-softmax logits will also be different even if their values after softmax are the same. Through experiments, we find using an appropriate distribution before softmax to initialize soft labels is crucial to maintaining the stability of the distillation. We have also tried to modify the distribution of logits without changing their values after softmax. However, we see this operation will greatly increase the instability of distillation. Moreover, we have tried to scale the values of logits during the initialization, which also leads to the degradation of performance.

### A.2.1 STABILITY

In the manuscript, we propose to generate `easy` and `hard` patterns sequentially to make the distillation stable enough for learning soft labels. The insight here is enabling the surrogate model to learn more `easy` patterns through the finite training steps and limited samples in inner optimization, such that the model can match the expert trajectories better. We find that simply increasing the update times in inner optimization can also stabilize the distillation. Because the surrogate model can learn `easy` patterns better through a longer learning process. However, increasing the update times in inner optimization will increase the memory requirement and the training cost. Thus we use the method introduced in the manuscript by default.

### A.2.2 SOFT LABEL OPTIMIZATION

We can choose to only replace original one-hot labels with soft labels but don't optimize them during the distillation. However, this will make the surrogate model harder to match the expert training trajectories, because the expert trajectories are trained with one-hot labels. As reflected in Figure 8 (Left), when soft labels are not optimized, the matching loss is higher than using one-hot labels. Although using unoptimized soft labels still performs better than one-hot labels because of the additional information contained in the soft labels, its performance is undermined by the under-matching.

The under-matching issue can be alleviated by optimizing soft labels during the distillation. As can be observed in Figure 8 (Left), when soft labels are optimized during the distillation, the matching loss becomes lower than using one-hot labels. Accordingly, the performance is improved, as can be observed in Figure 8 (Right).

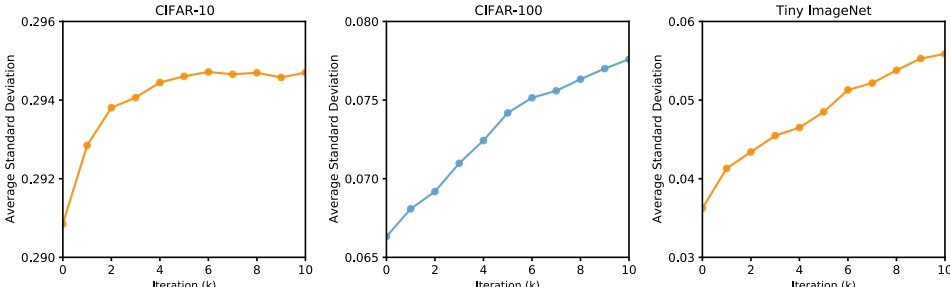

Figure 6: Visualization of the changing of soft labels in various datasets where IPC=50. The average standard deviation of soft labels tends to increase as the distillation goes on.

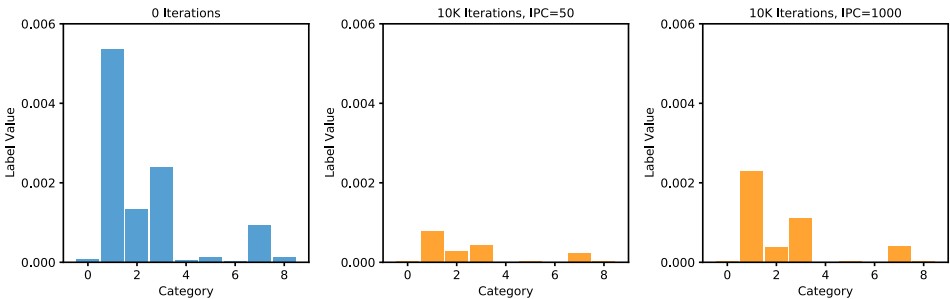

Figure 7: Visualization of the distribution of non-target soft labels of a synthetic image initialized with the same soft labels and image but distilled with different IPC settings. The values of non-target categories are suppressed more drastically when IPC is smaller.

## A.3 ABLATION ON SYNTHETIC STEPS

We find increasing the synthetic steps $N$ (Algorithm 1) will bring more performance gain when soft labels are utilized, as reflected in Figure 9 (Left). This is because the optimization of the surrogate model in the inner optimization affects how well it can match the expert trajectories. When soft labels are not utilized, the information contained in the synthetic dataset is relatively limited, thus the surrogate barely benefits from a longer learning process. Moreover, we find increasing the synthetic steps $N$ can also bring improvement in high IPC cases. As shown in Figure 9 (middle), setting $N$=80 performs best in the case with IPC=1000, where the batch size is set to 1000. In this case, in every iteration, the parameters of the surrogate model are optimized over 80K images (10K unduplicated images) contained in the synthetic dataset, while the target parameters are obtained by the optimization over 100k images (50k unduplicated images) contained in the original datasets. In this case, although the *length* of the training trajectory over the synthetic dataset $\mathcal{D}_{\mathrm{syn}}$ and the original dataset $\mathcal{D}_{\mathrm{real}}$ are similar, matching trajectories still can improve the training performance of the synthetic datasets. Based on this observation, we conjecture that the key to keeping TM-based methods effective is to ensure the number of unduplicated images contained in $\mathcal{D}_{\mathrm{syn}}$ is smaller than that of $\mathcal{D}_{\mathrm{real}}$, rather than use the *short* trajectory trained on $\mathcal{D}_{\mathrm{syn}}$ to match the *longer* one optimized over $\mathcal{D}_{\mathrm{real}}$.

## A.4 PREVIOUS TM-BASED METHODS IN LARGE IPC SETTINGS

We have tried to use previous TM-based methods to perform the distillation in larger IPC settings. The distillation logs of FTD (Du et al., 2023) are reported in Figure 9. As can be observed, FTD will undermine the training performance of the datasets in larger IPC cases. We have tried to tune its hyper-parameters including the learning rate, batch size, synthetic steps, and the upper bound of the sample range, but the effort can only slow down the rate of performance degradation it arose. Without aligning the difficulty of the generated patterns with the size of the synthetic datasets, previous TM-based methods can not keep being effective in high IPC settings.

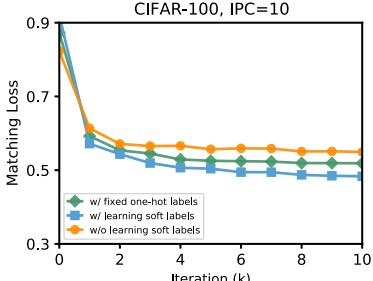 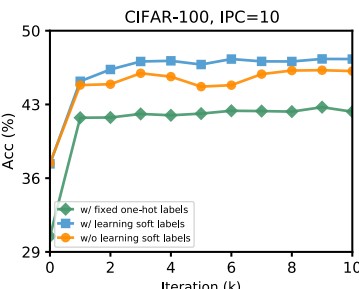

Figure 8: **Left**: Logs of the matching loss (smoothed with EMA), where the labels of the synthetic dataset are either one-hot labels, unoptimized soft labels, or soft labels that are optimized during the distillation. **Right**: Logs of performance of the distilled datasets. Learning soft labels during the distillation enables surrogate models to match expert trajectories better. Accordingly, its synthetic datasets have a better performance.

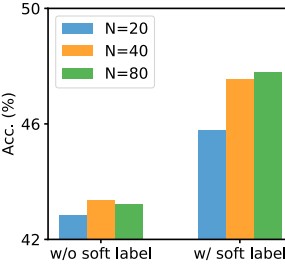 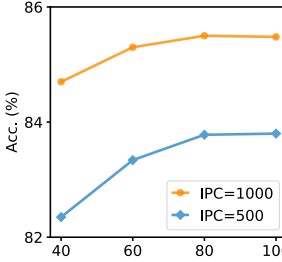 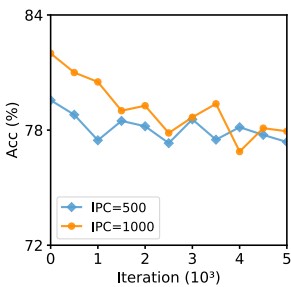

Figure 9: **Left**: (CIFAR-100, IPC=10) Ablation on synthetic step $N$, distillation with soft labels benefits more from a larger $N$. **Middle**: (CIFAR-10) Distillation in larger IPC settings can still benefit from a higher larger $N$. **Right**: Distillation log of FTD on CIFAR-10 with larger IPC. The performance of the synthetic datasets keeps being degraded as the distillation goes on.

## A.5    MORE DETAILED ANALYSIS ON MATCHING LOSS

Here we provide more results and additional analysis about the matching loss over expert trajectories. As can be observed in Figure 10, initially, the matching loss over the former part of the trajectories is always lower than the one over the later part. This indicates earlier trajectories are relatively easier to match compared with the later ones. In other words, the patterns that surrogate models need to learn to match the early trajectories are relatively easier.

Moreover, it is interesting to observe that matching later trajectories can also reduce the matching loss over early trajectories in high IPC cases. We hypothesize that this is because, in the late training phases, DNNs just *prefer* to learn `hard` patterns to help identify the outliers, while a few `easy` patterns are also learned in late training phases. From this perspective, TM-based methods might not be the most efficient way to distill datasets with large IPC, since matching later trajectories still will generate a few `easy` patterns.

We can also find that when IPC is small, matching late trajectories will raise the matching loss over the early trajectories. This indicates generating `hard` patterns is harmful for the model to learn basic (`easy`) patterns to obtain the basic capacity to perform the classification when data is limited. This coincide with the observation in the *dataset pruning* area: preserving `hard` samples perform worse when only limited samples are kept (Sorscher et al., 2022).

## A.6    GUIDANCE FOR ALIGNING DIFFICULTY

Our difficulty alignment aims at letting the models trained on the synthetic dataset learn as many `hard` patterns as possible, without compromising their capacity to classify `easy` patterns. For TM-based methods, this can be quantified by the matching loss over a distillation-uninvolved expert trajectory, as we have analyzed in section A.5. Specifically, we want to add patterns that can help to **reduce** the matching loss over the latter part of the expert trajectory **without increasing** the matching

---

**Algorithm 1:** Pipeline of our method

---

**Input:** $\{\tau^*\}$: set of expert parameter trajectories. $N$: update times of the surrogate network in
each inner optimization. $M$: update times between the start and target expert parameters.
$T^-, T, T^+$: lower, current upper, final upper bound of the sample range of $t$. $\mathcal{D}_{\text{real}}$: original
dataset. $I$: interval for expanding the sampling range.
Sample a model $f_{\theta^*}$ from $\{\tau^*\}$.
Construct $\mathcal{D}_{\text{sub}} = \{(x_i, \text{softmax}(L_i))|(x_i, y_i) \in \mathcal{D}_{\text{real}} \textbf{ and } \text{argmax}(L_i) == y_i\}$, where
$L_i = f_{\theta^*}(x_i)$.
Randomly sample data from $\mathcal{D}_{\text{sub}}$ to initialize synthetic dataset $\mathcal{D}_{\text{syn}}$.
**for** iteration $\leftarrow 0$ *to* max_iteration **do**

    Randomly sample an expert training trajectory $\tau^* \in \{\tau^*\}$ with $\tau^* = \{\theta_i^*\}_0^n$
    Select random start timestamp $t$, where $T^- \leq t \leq T$
    Sample $\theta_t^*, \theta_{t+M}^*$ from $\tau^*$, initialize $\hat{\theta}_t = \theta_t^*$
    **for** $i \leftarrow 0$ *to* $N$-1 **do**
        $b_{t+i} \sim \mathcal{D}_{\text{syn}}$               $\triangleright$ sample a mini-batch of distilled dataset
        $\hat{\theta}_{t+i+1} = \hat{\theta}_{t+i} - \alpha\nabla\ell(\hat{\theta}_{t+i}, b_{t+i})$      $\triangleright$ update surrogate model with CE loss
    Compute matching loss between $\hat{\theta}_{t+N}$ and $\theta_{t+M}^*$ with Eq. 1
    Update $(x_i, \text{softmax}(L_i)) \in \{b\}_t^{t+N-1}$ and $\alpha$ with respect to the matching loss
    **if** (iteration % $I == 0$ ) **and** ($T < T^+$) **then**
         $T = T + 1$

**Output:** distilled dataset $\mathcal{D}_{\text{syn}}$ and learning rate $\alpha$

---

loss over the former part. In practice, we realize it by only matching a certain part of the expert
trajectories during the distillation.

Here we introduce how to tune the values of $T^-$, $T$, and $T^+$ (Algorithm 1). For an initialized
synthetic dataset $\mathcal{D}_{\text{syn}}$, we first set $T^- = 0$ and $T^+ = T^- + 10$ to perform the distillation for
50 iterations, where the matching loss over a distillation-uninvolved expert trajectory is recorded.
Then we simultaneously increase the values of $T^-$ and $T^+$ until the distillation will not increase the
matching loss over the latter part of the expert trajectory. Subsequently, we increase the value of
$T^+$ until the distillation will increase the matching loss over the former part of the expert trajectory.
After deciding the values of $T^-$ and $T^+$, we let them respectively be the lower- and upper-bound
of the sample range and perform the distillation. During the distillation, we record the value of $t$
if the matching loss is larger than 1, which denotes the surrogate model can not match the expert
trajectory. Then $T$ is set as the minimum recorded value, to avoid matching too hard trajectories in
the beginning.

## A.7 MORE RELATED WORK

Two early works (Bohdal et al., 2020; Sucholutsky & Schonlau, 2021b) in the dataset distillation
area also focus on optimizing the labels of the datasets. Specifically, based on the dataset distillation
algorithm proposed by Wang et al. (2018), Sucholutsky & Schonlau (2021b) propose to optimize
labels during the distillation, while Bohdal et al. (2020) choose to only distill soft labels without
optimizing the training data. Different from them, we use the pre-trained model to initialize soft
labels, which contain more information. Furthermore, our method is based on matching training
trajectories (Cazenavette et al., 2022) rather than the method proposed by Wang et al. (2018).

Recently, several methods are proposed to improve the performance, efficiency and suitability of
dataset distillation. For example, Liu et al. (2022) proposed to use hallucinations to enlarge the
synthetic datasets in the deployment stage. Subsequently, Wang et al. (2023) achieved this goal by
distilling the target dataset into a generative model. Moreover, (Zhang et al., 2023; Liu et al., 2023b)
were proposed to accelerate the distillation and Liu et al. (2023a) proposed a method that allows
adjusting the size of the distilled dataset during the deployment stage. Recently, Chen et al. (2024)
proposed to improve the quality of the synthetic dataset with a carefully designed distillation schedule.
Moreover, dataset distillation has also been successfully applied in condensing gragh data (Jin et al.,
2022; Yang et al., 2023; Zhang et al., 2024a;b) and multi-modality data (Wu et al., 2023).

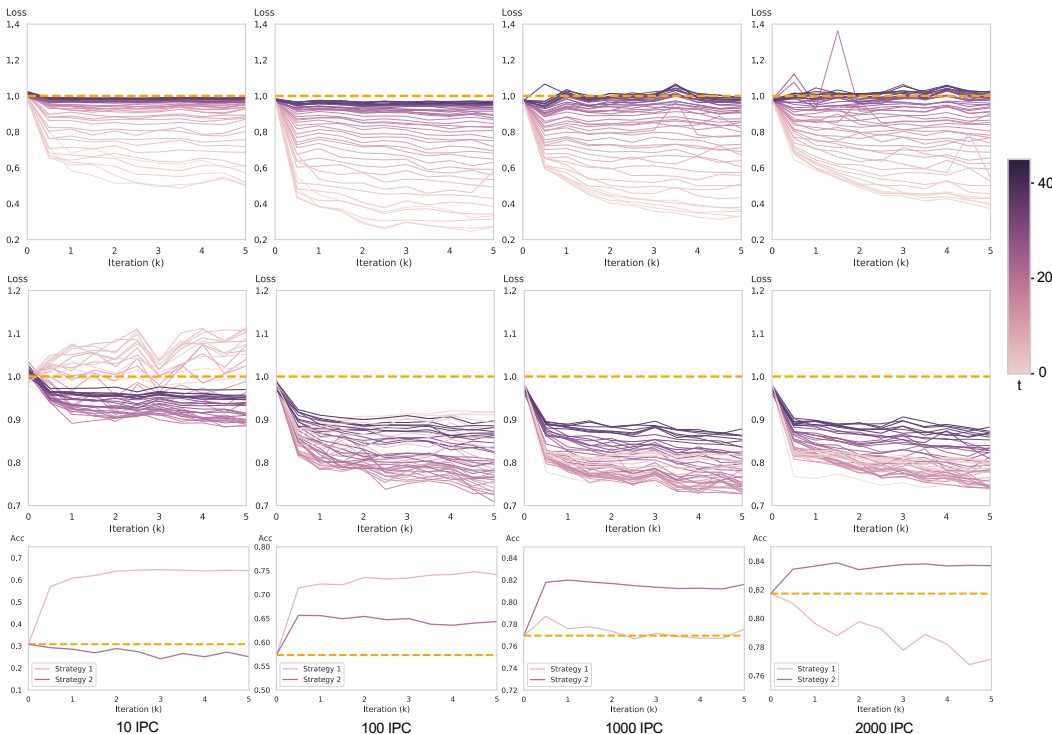

Figure 10: More detailed results of experiments reported in Figure 2. We train the expert models on CIFAR-10 for 40 epochs. Then we distill datasets with two strategies: (1) matching the early part of expert training trajectories, where $\theta_t^* \in \{\theta_0^*...\theta_{20}^*\}$. (2) matching the latter part of expert training trajectories, where $\theta_t^* \in \{\theta_{20}^*...\theta_{40}^*\}$. The first row shows the matching loss over a distillation-uninvolved expert trajectory, where the distillation is performed with strategy 1, and the second row shows the loss of strategy 2. In the first two rows, lines with darker color indicates the matching loss over more later part of the trajectories. $t$ denotes the timestamp of the start parameters used for matching (Algorithm 1). The last row shows the performance of the datasets distilled by different strategies with various IPC settings.

## A.8 SETTINGS

**Distillation.** Consistent with previous works (Cazenavette et al., 2022; Du et al., 2023), we perform the distillation for 10000 iterations to make sure the optimization is fully converged. We use ZCA whitening in all the involved experiments by default.

**Evaluation.** We keep our evaluation process consistent with previous works (Cazenavette et al., 2022; Du et al., 2023). Specifically, we train a randomly initialized network on the distilled dataset and then evaluate its performance on the entire validation set of the original dataset. Following previous works (Cazenavette et al., 2022; Du et al., 2023), the evaluation networks are trained for 1000 epochs to make sure the optimization is fully converged. All the results are the average over five trials. For fairness, experimental results of previous distillation methods in low IPC settings are obtained from (Du et al., 2023), while their results in high IPC cases come from (Cui et al., 2022).

Since the exponential moving average (EMA) used in FTD (Du et al., 2023) is a plug-and-play technique that hasn't been utilized by previous matching-based methods, for a fair comparison, we reproduce FTD with the official released code without using EMA. Accordingly, we do not use EMA in our method.

**Network.** We use various networks to evaluate the generalizability of our distilled datasets. Specifically, to scale ResNet, LeNet, and AlexNet to Tiny-ImageNet, we increase the stride of their first convolution layer from 1 to 2. For VGG, we increase the stride of its last max pooling layer from 1 to 2. The MLP utilized in our evaluation has one hidden layer with 128 units.

**Hyper-parameters**. We report the hyper-parameters of our method in Table 6. Additionally, for all the experiments with optimizing soft labels, we set its momentum to 0.9. We find learning labels with a low momentum will somewhat increase the instability of the distillation. We conjecture this is because the optimized soft labels are easy to overfit the expert trajectories considering we only match one trajectory in each iteration.

**Compute resources.** Our experiments are run on 4 NVIDIA A100 GPUs, each with 80 GB of memory. The heavy reliance on GPU memory can be alleviated by TESLA (Cui et al., 2023) or simply reducing the synthetic steps $N$, which will not cause too much performance degradation. For example, reducing the synthetic steps $N$ from 80 to 40 saves about half the GPU memory, while it only makes the performance drop by around 0.8% for CIFAR-10 with IPC=1000, 0.7% for CIFAR-100 with IPC=100, and 0.4% for TinyImageNet with IPC=50.

| Dataset | IPC | N | M | $T^-$ | $T$ | $T^+$ | Interval | Synthetic Batch Size | Learning Rate (Label) | Learning Rate (Pixels) |
|---|---|---|---|---|---|---|---|---|---|---|
| CIFAR-10 | 1 | 80 | 2 | 0 | 4 | 4 | - | 10 | 5 | 100 |
|  | 10 | 80 | 2 | 0 | 10 | 20 | 100 | 100 | 2 | 100 |
|  | 50 | 80 | 2 | 0 | 20 | 40 | 100 | 500 | 2 | 1000 |
|  | 500 | 80 | 2 | 40 | 60 | 60 | - | 1000 | 10 | 50 |
|  | 1000 | 80 | 2 | 40 | 60 | 60 | - | 1000 | 10 | 50 |
| CIFAR-100 | 1 | 40 | 3 | 0 | 10 | 20 | 100 | 100 | 10 | 1000 |
|  | 10 | 80 | 2 | 0 | 30 | 50 | 100 | 1000 | 10 | 1000 |
|  | 50 | 80 | 2 | 20 | 70 | 70 | - | 1000 | 10 | 1000 |
|  | 100 | 80 | 2 | 30 | 70 | 70 | - | 1000 | 10 | 50 |
| TI | 1 | 60 | 2 | 0 | 15 | 20 | 400 | 200 | 10 | 10000 |
|  | 10 | 60 | 2 | 10 | 50 | 50 | - | 250 | 10 | 100 |
|  | 50 | 80 | 2 | 40 | 70 | 70 | - | 250 | 10 | 100 |

Table 6: Hyper-parameters for different datasets.

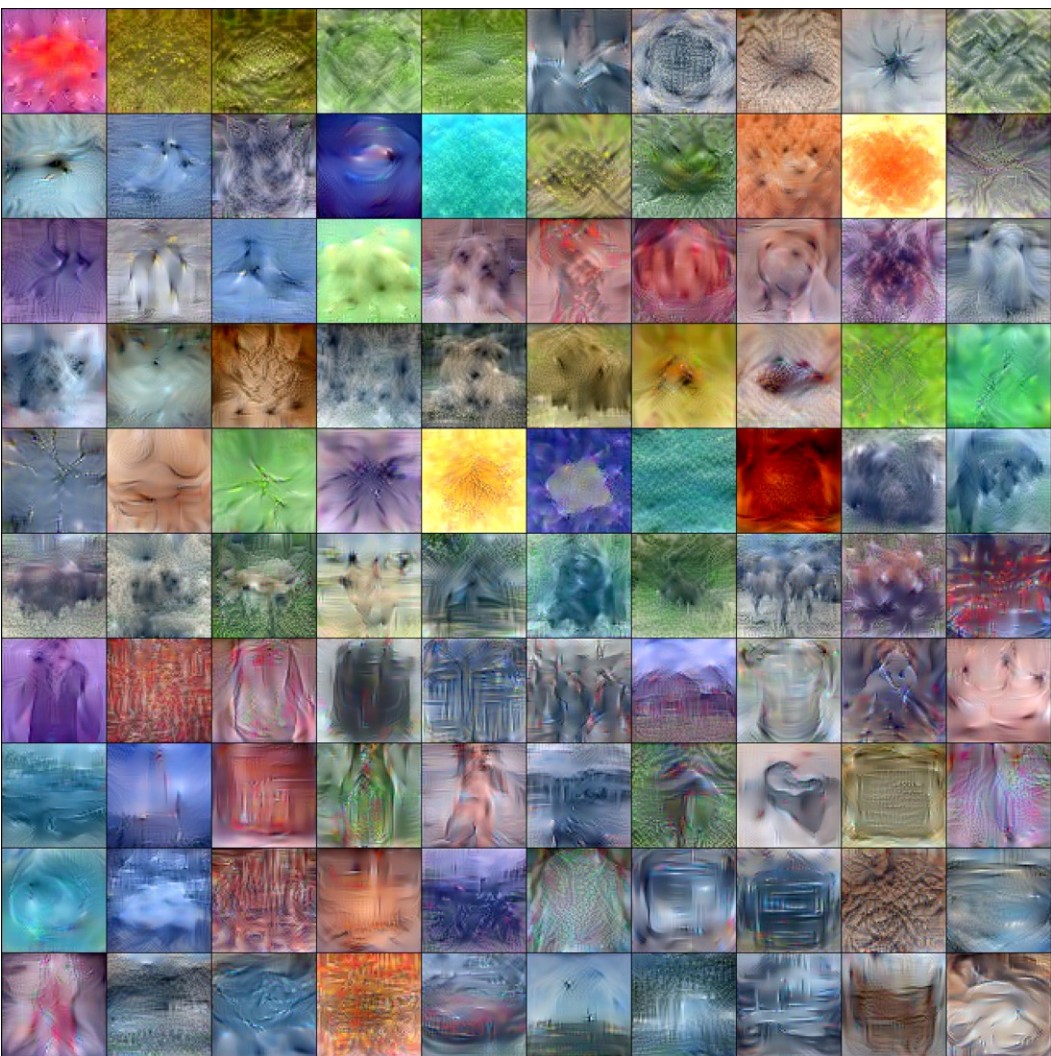

Figure 11: (Tiny ImageNet, IPC=1) Visualization of distilled images (1/2).

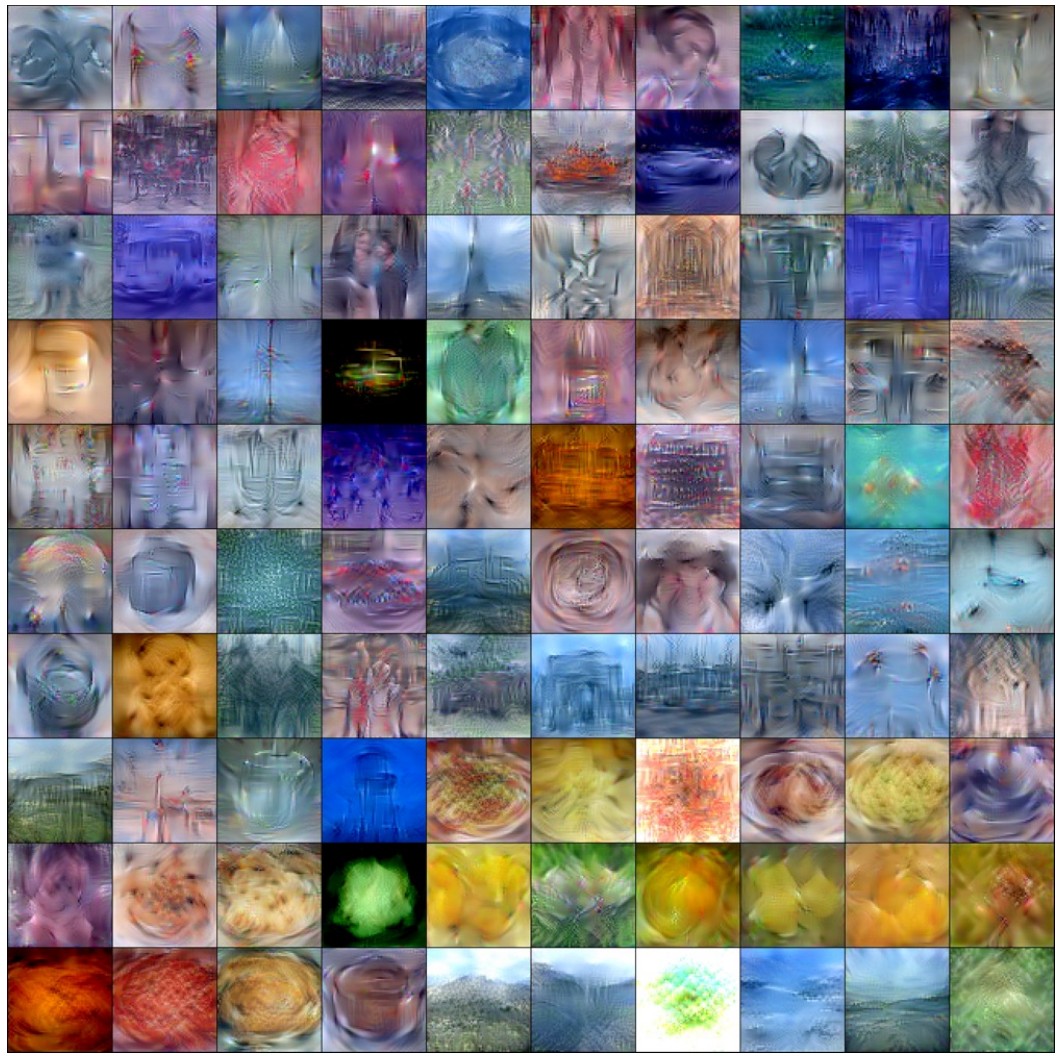

Figure 12: (Tiny ImageNet, IPC=1) Visualization of distilled images (2/2).

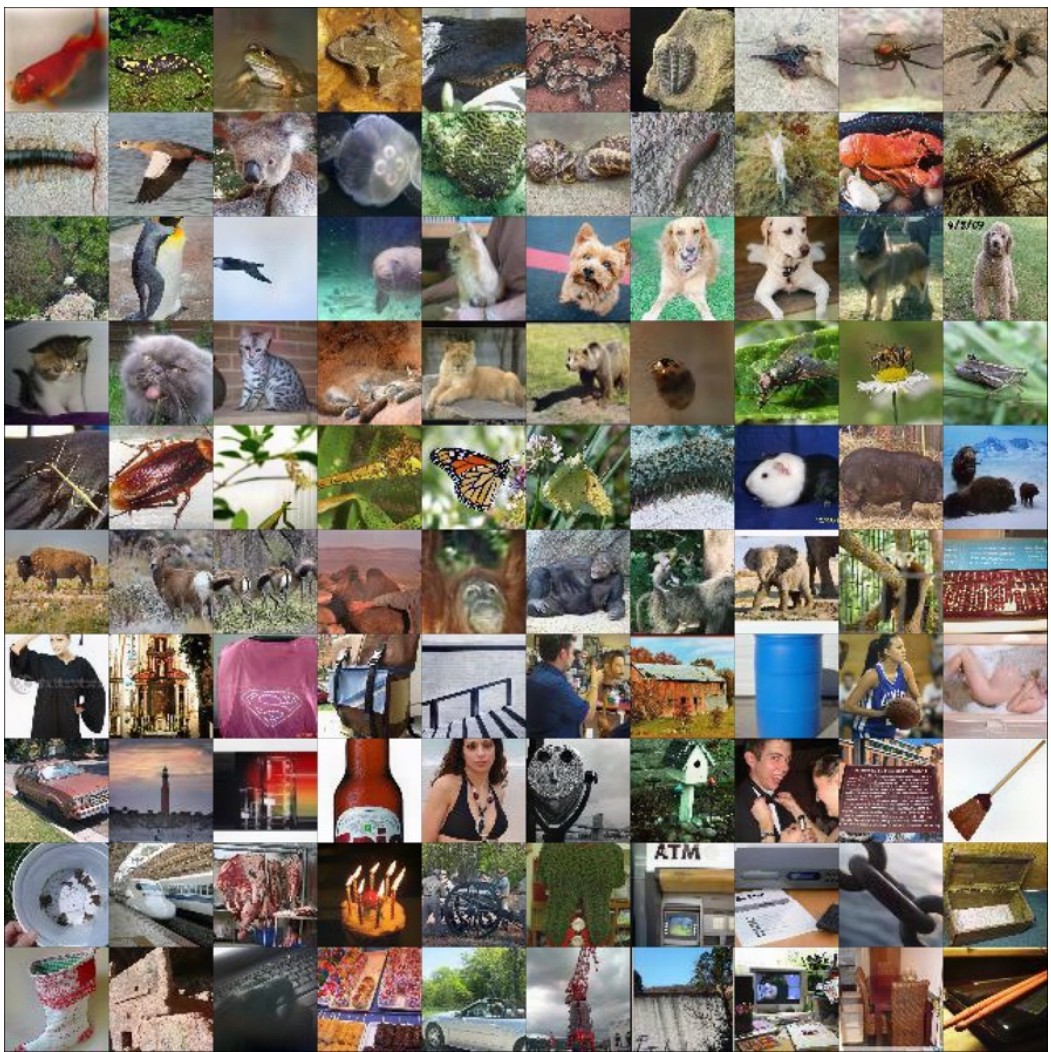

Figure 13: (Tiny ImageNet, IPC=50) Visualization of distilled images (1/2).

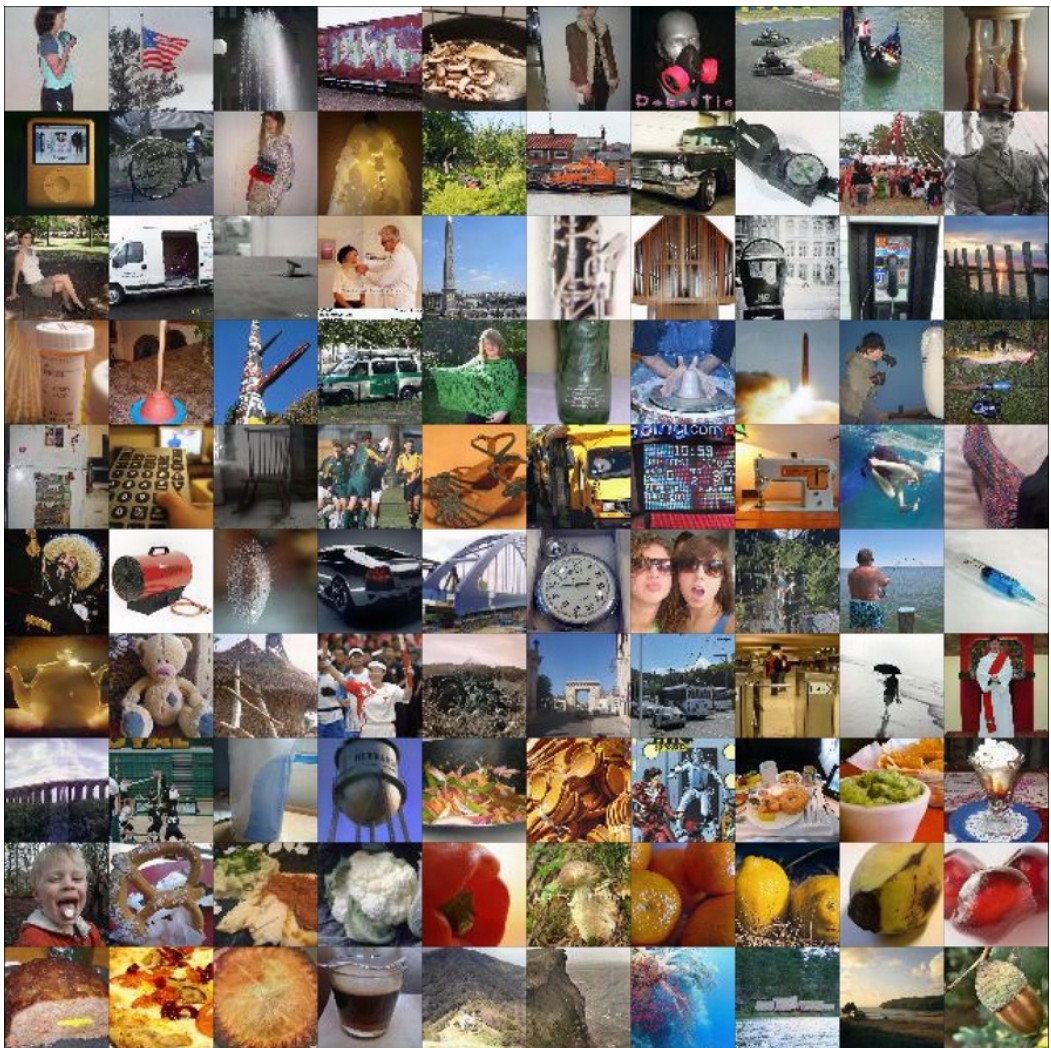

Figure 14: (Tiny ImageNet, IPC=50) Visualization of distilled images (2/2).

