# OpenReview forum: "Towards Lossless Dataset Distillation via Difficulty-Aligned Trajectory Matching"
_ICLR.cc/2024/Conference — ICLR 2024 poster_

### Official Review · Reviewer_VsZC · 2023-10-29

**Soundness:** 3 good
**Presentation:** 3 good
**Contribution:** 3 good
**Rating:** 8
**Confidence:** 4

**Summary:**

The papers proposes a new method to perform dataset distillation under large IPCs. The authors made server observations regarding the trajectory matching process and found that easy patterns are learnt at early states and hard patterns are learnt at later stages. The authors also found that learning labels will help boost the distillation performances. Through gradually increasing the limit of matched expert trajectory epochs, the method achieves SOTA performances on large IPCs. Especially, when IPC increases to 500 or 1000 where all previous methods fail to perform better than random baseline, the proposed method is the first to be able to perform equivalently well as training on the whole dataset or better. The proposed method also achieves SOTA results on cross-architecture evaluations.

**Strengths:**

1. the method solves the problem that previous DD methods failed to solve: when IPC becomes large such as 500, their performances start to degrade to similar levels as random selection.
2. The paper is well written and easy to follow.
3. The paper introduced the observations gradually and solved the problems through carefully designed strategies.
4. The comparison to baseline methods are extensive.
5. The method is the first to scale to large IPCs such as 500, 1000 and achieves lossless performance on the 3 datasets.
6. The paper also showed lots of insights for the reason community to gain a better understanding of Matching Training Trajectories(MTT) which is one of the recently proposed SOTA methods.

**Weaknesses:**

1. In table 1, can the authors indicate which DATM's results are achieved with label learning and which are not? Performance-wise, DATM's performance is very similar to other MTT based methods such as MTT itself with small IPCs, if soft label is used, does that mean DATM will cause MTT's performance to drop under smaller IPCs which is then compensated by soft label learning?

2. Although mentioned briefly in 5.2, can the authors also specify the training resources such as number of GPUs and their type and the training cost in either 4.1 or appendix. This will provide a guidance for other researchers, especially on large IPCs and datasets. This leads to my next question

3. Does this 1.05 times higher cost apply to all settings or it's just for CIFAR-10 (Sec 5.2 mentioned it's for CIFAR-10)? How about the extra training cost on other datasets?

4. Some claims conflicts with the settings adapted by the method, e.g. in section 4.3, the authors first mention that "We use logits generated by the pre-trained model to initialize soft labels", then later in the same section, the authors claim again "This is because using soft labels without optimizing them will enlarge the discrepancy between the training trajectories over the synthetic dataset and the original one, **considering the experts are trained with one-hot labels**". If the later claim (considering the experts are trained with one-hot labels) is correct, shouldn't initialize the soft labels with a distribution close to one-hot have a smaller gap with expert training trajectories during the matching phase? The decision to use pre-trained model to initialize soft labels can be understood, however I think the explanation given later is confusing

5. Have the authors tried to increase $T^{-}$ as well instead of just increasing the current upper?

**Questions:**

See my questions above inside weakness.

---

> ### Author Response · Authors · 2023-11-16
> **Response to Reviewer VsZC (1/2)**
>
> > Q1. Which DATM's results are achieved with label learning? In low IPC settings, DATM's performance is similar to MTT. Will DATM perform worse than MTT without the utilization of soft labels when IPCs are small?
>
> We thank the reviewer for the constructive feedback, which helps to improve the clarity of our paper. We answer the questions one by one as follows. Hope it can address the reviewer’s concern.
>
> + All results in Table 1 are achieved with label learning.
> + In low IPC settings, DATM will not cause MTT's performance to drop when soft labels are not utilized.
> + DATM will have performance similar to previous TM-based methods only when **both** IPC and the number of categories contained in the classification mission are small, because:
> 	1. DATM's performance improvement comes from 'difficulty alignment' and 'learning soft labels'.
> 	2. Since previous methods correctly choose to match only early (easy) trajectories when IPC is small, our difficulty alignment strategy will not bring much improvement compared with previous works. In this case, most improvement of DATM comes from label learning.
> 	3. However, soft labels can store only a little information when the amount of the categories contained in the classification mission is limited; hence, DATM only outperforms FTD by 0.9% on CIFAR-10 (ipc=1, 10 classes) but the figure increased to 3.5% for CIFAR-100 (ipc=1, 100 classes) and 6.6% for TinyImageNet (ipc=1, 200 classes).
>
> ---
>
> > Q2. More details about training resources.
>
> Our experiments are run on 4 NVIDIA A100 GPUs, each with 80 GB of memory. We consume large GPU memory because we find increasing synthetic steps $N$ helps to improve the performance when soft labels are utilized (section A.3).
>
> The heavy reliance on GPU memory can be alleviated by TESLA [1] or simply reducing the synthetic steps $N$, which will not cause too much performance drop. For example, reducing the synthetic steps $N$ from 80 to 40 saves nearly half the GPU memory, while only making the performance drop by 0.8% for CIFAR-10 with IPC=1000, 0.7% for CIFAR-100 with IPC=100, and 0.4% for TinyImageNet with IPC=50.
>
> **Section A.8 is now updated to include the above contents.**
>
> [1] Scaling Up Dataset Distillation to ImageNet-1K with Constant Memory, ICML 2023
>
> ---
>
> > Q3. More details about the distillation cost.
>
> We check the distillation logs and report the training costs as follows (evaluation time is not counted):
>
> + **Training costs in low IPC cases**
>
> |                  |CIFAR-10 |CIFAR-100 | TinyImageNet |
> |:----------------:|:---------:|:---------:|:------------:|
> |       IPC        |10 |   10    |      10      |
> | Converge at Iter |7000|   4500    |    7000    |
> | GPU hrs/1k iter  |1.3 |   3.2    |    3.56    |
> | Total GPU hrs    |9.1 |   14.4     |       24.92       |
>
> + **Training costs in high IPC cases**
>
> |                  |CIFAR-10 |CIFAR-100 | TinyImageNet |
> |:----------------:|:---------:|:---------:|:------------:|
> |       IPC        |1000 |   100    |      50      |
> | Converge at Iter |3000|   5000    |    6000    |
> | GPU hrs/1k iter  |3.2 |   3.2    |    3.89    |
> | Total GPU hrs    |9.6 |   16     |       23.34       |
>
> **Conclusion**
> Overall, **the distillation costs don't increase significantly as IPC grows**.
>
> ---
>
> > Q4.  Questions about the motivation for optimizing soft labels
>
> We apologize for any confusion; we hope to clarify our intentions here.
>
> **Why we don't use one-hot labels to initialize the soft labels ?**
> We choose to optimize the pre-softmax logits because it is unclear to us how to initialize the pre-softmax logits such that the post-activations will be similar to a one-hot vector (in a way that would be easy to optimize).
>
> **Why do we claim using unoptimized soft labels enlarges the discrepancy between trajectories?**
> To support our claim better, Figure 8 is updated to report the matching loss and performance of the distillations performed with one-hot labels, unoptimized soft labels, and optimized soft labels. For convenience, we summarize Figure 8 here:
> 1. Matching loss: unoptimized soft labels > one-hot labels > optimized soft labels.
> 2. Performance: optimized soft labels > unoptimized soft labels > one-hot labels.
>
> Analysis:
> + According to the matching loss, we can infer that, although the performance is improved due to the additional information stored in soft labels, replacing one-hot labels with unoptimized soft labels will make it harder for the surrogate model to match the expert trajectories.
> + By optimizing soft labels during the distillation, this under-matching issue is alleviated. Accordingly, the performance is improved.
>
> **Section A.2.2 is updated to introduce the above contents.**

---

> ### Author Response · Authors · 2023-11-16
> **Response to Reviewer VsZC (2/2)**
>
> >Q5. Have the authors tried to increase $T^-$ as well instead of just increasing the current upper?
>
> Yes, we have tried this. The insight behind this operation is: considering the synthetic data needs to contain enough 'easy' patterns to support 'hard' patterns, maybe we only need to generate 'easy' patterns in the beginning (i.e., maybe we can stop matching early trajectories after enough 'easy' patterns are generated).
>
> In practice, we tried this on CIFAR-10 with IPC=50 where the default matching range is [$T^-$, $T$] ( Initially $T^-=0$ and $T=20$; $T$ is gradually increased to 40 during the distillation). We tried to increase $T^-$ simultaneously with $T$, which means that the matching range is a 'sliding window' with a length 20.
>
> The results are worse than fixing $T^-=0$. We think this is caused by catastrophic forgetting: matching only later trajectories in the late distillation phase lets the synthetic images forget the 'easy' patterns learned by matching early trajectories.

---

> ### Author Response · Authors · 2023-11-20
> **Further Discussions with Reviewer VsZC**
>
> Dear Reviewer VsZC:
>
> Thank you so much again for your time and efforts in assessing our paper. Hope our response has addressed your concerns. We would be happy to have further discussions if you have any other questions or suggestions. Thanks for helping improve our paper!
>
> Best,
> Authors from submission 243

---

> > ### Comment · Reviewer_VsZC · 2023-11-22
> > **Thank authors for the response**
> >
> > Thank the authors for the effort clarifying my confusions. The responses align with my expectations. Although the algorithm novelty is limited, the findings and empirical results are important and the new setting (Lossless DD) introduced is also a big contribution to the field of dataset distillation. Therefore I will keep my score: 8 and recommend acceptance of the paper.

---

> > > ### Author Response · Authors · 2023-11-22
> > > **Response to Reviewer VsZC**
> > >
> > > Thanks again for the acknowledgment and the efforts in assessing this work. We will keep exploring this important topic.

---

### Official Review · Reviewer_G6x3 · 2023-10-30

**Soundness:** 3 good
**Presentation:** 3 good
**Contribution:** 3 good
**Rating:** 6
**Confidence:** 3

**Summary:**

This work aims to achieve lossless dataset distillation by synthesizing a small synthetic dataset such that a model trained on this synthetic set will perform equally well as a model trained on the full, real dataset. Concretely, the author identified the problem of performance not being maintained as the size of the synthetic dataset grows. The authors propose a way to process early trajectories (easy patterns) and late ones (hard patterns) separately.

**Strengths:**

1) The author identified an important problem from an interesting perspective. The current dataset distillation algorithms cannot handle it well.

2) The model performance on the synthesized dataset is better than that on the dataset synthesized by other dataset distillation methods.

**Weaknesses:**

1. The goal of this work is to achieve lossless dataset distillation. This requires a formal definition, what is lossless dataset distillation exactly?  Do models achieve the same performance on the original dataset as on the synthesized one?

2. Why the model performance on the synthesized one is better than the one on the full dataset? Does that mean that the proposed whole dataset distillation works like a regularization method somehow?

3. If a synthesized dataset is indeed lossless, a model with a totally different architecture can also perform well on the synthesized dataset. There lack of such experiments.

4. This work improves previous work when the size of the synthesized dataset becomes larger. The lossless dataset distillation is an overclaim to me.

**Questions:**

1) Please explain why the performance across model architecture is so low.

2) Please see my statements in weakness.

---

> ### Author Response · Authors · 2023-11-16
> **Response to Reviewer G6x3 (1/2)**
>
> > Weakness 1,3,4: Question about 'lossless dataset distillation'.
>
> We thank the reviewer for the constructive feedback, which helps to improve the clarity of our paper. We answer the questions one by one as follows. We hope our responses help alleviate the reviewer’s concerns.
>
> **Definition**
> By “lossless dataset distillation”, we mean distilling a small synthetic dataset such that a network trained on this dataset reaches the same test performance as a network trained on the full, real dataset.
>
> **Limitation of our work**
> In this work, we achieve lossless dataset distillation on ConvNet, the “backbone” architecture used for distillation. So far, we haven't reached the ultimate goal of dataset distillation: distilling a dataset that achieves lossless performance across all architectures. However, we think our 'difficulty alignment strategy' is very important for achieving this ultimate goal, so the title of our paper is "**Towards** Lossless Dataset Distillation via Difficulty-Aligned Trajectory Matching".
>
> To avoid misunderstanding, **we add the limitation analysis of our work in section 5.4 and update our submission PDF accordingly**. For convenience, we show section 5.4 (newly added) as follows.
>
> ''In this work, we achieve lossless dataset distillation on ConvNet, the backbone used for distillation. However, when evaluating the distilled datasets with other models, the performance drops still exist, possibly due to the fact that models with different capacities need varying amounts of training data. How to overcome this issue is still a challenging problem. In addition, it is hard to scale TM-based methods to large-scale datasets due to the high distillation cost. How to generate hard patterns in a more efficient way would be the goal of our future work.''
>
> ---
>
> > Weakness 2: Why the model trained on the synthesized dataset can perform better than the one trained on the full dataset?
>
> This is because some samples contained in the original dataset are redundant for DNNs. Recent dataset reduction studies have shown that, with a well-designed strategy, the reduced dataset can have better performance than the original, full dataset [1][2][3]. Accordingly, with a good distillation algorithm, we can also distill a dataset that is better than the full dataset.
>
> On the other hand, considering stronger networks are more hungry for training data, datasets distilled by ConvNet may still be redundant for weaker networks while being less informative for stronger models. Based on this, we think there is still a long way to go to distill datasets that are lossless for any network.
>
> [1] Deep learning on a data diet: Finding important examples early in training, NeurIPS 2021.
> [2] An Empirical Study of Example Forgetting during Deep Neural Network Learning, ICLR 2019.
> [3] Too Large; Data Reduction for Vision-Language Pre-Training, ICCV 2023.

---

> ### Author Response · Authors · 2023-11-16
> **Response to Reviewer G6x3 (2/2)**
>
> > Q1: Question about cross-architecture generalization.
>
> Cross-architecture generalizability is an important metric for evaluating distillation algorithms, according to previous works [1, 2], it is performed by:
> 1. Perform the distillation using the backbone network (ConvNet).
> 2. Train networks with different architectures from scratch using distilled dataset.
> 3. Evaluate trained networks' performance on the test set of the target dataset.
>
> Since the distillation is performed with a single type of network (ConvNet), distilled datasets will inevitably be biased toward ConvNet. As shown in the table below (data comes from Table 2 (a)), ConvNet benefits the most from the distillation. Compared with previous methods, **our synthetic dataset has significantly better generalizability in low IPC settings**, achieving state-of-the-art performance.
>
> |  Method  |  ConvNet  | ResNet 18 |    VGG    |  AlexNet  |
> |:--------:|:---------:|:---------:|:---------:|:---------:|
> |  Random  |   33.46   |   31.95   |   32.18   |   26.65   |
> |   MTT    |   45.68   |   42.56   |   41.22   |   40.29   |
> |   FTD    |   48.90   |   46.65   |   43.24   |   42.20   |
> | **Ours** | **55.03** | **51.71** | **45.38** | **45.74** |
>
> Note: experiments performed on CIFAR-100 with IPC=50.
>
> According to the above table, the performance differences may also be attributed to the mismatch between model capacity and compressed dataset size. Therefore, we conducted additional experiments with larger IPC, and the results are shown in Table 4 (for convenience, parts of them are shown below).
>
> + **CIFAR-10, data keep ratio: 20%, IPC=1000**
>
> | Method      | ConvNet | ResNet18 | VGG11 | AlexNet |
> |:-------------:|:---------:|:----------:|:-------:|:---------:|
> | Random      | 78.38   | 84.58    | 80.81 | 80.75   |
> | Glister     | 62.46   | 81.10    | 78.07 | 70.55   |
> | Forgetting  | 76.27   | 85.67    | 82.04 | 81.35   |
> | **DATM**    | **85.50** | **87.22** | **84.65** | **85.14**   |
>
> ---
>
> + **CIFAR-100, data keep ratio: 20%, IPC=100**
>
> | Method      | ConvNet | ResNet18 | VGG11 | AlexNet |
> |:-------------:|:---------:|:----------:|:-------:|:---------:|
> | Random      | 42.80   | 47.48    | 42.69 | 38.05   |
> | Glister     | 35.45   | 42.49    | 43.06 | 28.58   |
> | Forgetting  | 45.52   | 51.44    | 43.28 | 43.47   |
> | **DATM**    | **57.50** | **57.98** | **55.10** | **55.69**   |
>
> ---
>
> + **Tiny ImageNet, data keep ratio: 10%, IPC=50**
>
> | Method| ConvNet | ResNet18 | VGG11 | AlexNet |
> |:------:|:---------:|:----------:|:-------:|:---------:|
> | Random      | 15.00   | 17.73    | 22.51 | 14.03   |
> | Glister     | 17.32   | 18.84    | 19.10 | 11.68   |
> | Forgetting  | 20.04   | 19.38    | 23.77 | 12.13   |
> | **DATM**    | **39.68** | **36.12** | **38.35** | **35.10**   |
>
> Note:  According to [3, 4], previous distillation methods perform similar or even worse than random selection when IPC is high, we choose to compare DATM with core-set selection methods [5,6], which is a stronger baseline in high IPC cases.
>
> **Conclusion**
> As can be observed, our methods generalize well on unseen architectures, performing better than core-set selection methods. Moreover, ResNet 18 performs better than ConvNet in the majority of cases, indicating a significant improvement in generalization under a larger IPC.
>
> **Future plans**
> The cross-architecture generalizability is very important for dataset distillation, we will further explore how to improve it by:
> 1. Using a model pool (contains various networks) as the backbone to perform the distillation.
> 2. Applying our method on GLad [7], to see if a generator prior can help to improve the generalizability in high IPC cases.
>
> [1] Dataset Condensation with Gradient Matching, ICLR 2021.
> [2] Dataset Distillation by Matching Training Trajectories, CVPR 2022.
> [3] DC-BENCH: Dataset Condensation Benchmark, NeurIPS 2022.
> [4] Dataset Quantization, ICCV 2023.
> [5] Glister: Generalization based data subset selection for efficient and robust learning, AAAI 2021.
> [6] An empirical study of example forgetting during deep neural network learning, ICLR 2018.
> [7] Generalizing Dataset Distillation via Deep Generative Prior, CVPR 2023.

---

> ### Author Response · Authors · 2023-11-20
> **Further Discussions with Reviewer G6x3**
>
> Dear reviewer G6x3:
>
> Thanks again for your valuable feedback, which helps to improve the clarity of our paper. According to your constructive comments, we made the necessary adjustments to introduce the limitations of our work in Section 5.4.
>
> As the rebuttal period is about to close, may I know if our rebuttal addresses your concerns? Thank you for taking the time to review our work and provide your insightful comments.
>
> Best,
> Authors from submission 243

---

> ### Author Response · Authors · 2023-11-21
> **Summary of Response**
>
> Dear Reviewer G6x3,
>
> Considering the limited time available, and in order to save the reviewer's time, we summarize our responses here.
>
>  1. **[Questions about 'lossless dataset distillation']**
> 	By 'lossless dataset distillation', we mean distilling a small synthetic dataset such that a network trained on this dataset reaches the same test performance as a network trained on the full, real dataset.
> 	In this work, we achieve lossless performance for the distillation backbone (ConvNet). However, when evaluating with other networks, the performance drops still exist. To avoid misunderstandings, **we add the limitation analysis of our work in section 5.4 and update our submission PDF accordingly**.
>
>  2. **[Question about cross-architecture generalization.]**
> 	**Our method generalizes well in both low and high IPC cases**. We compare the generalizability in both low and high IPC settings (results are reported in Table 2 (a) and Table 4).  Results show our method performs best in every setting, reflecting its outstanding generalizability.
>
> 3. **[Why synthesized dataset can perform better than full dataset?]**
> Because some samples contained in the full dataset are redundant for DNNs. Previous dataset pruning methods have shown that, with a good pruning strategy, pruned datasets can perform better than the full datasets. Accordingly, with a well-designed distillation strategy, synthesized datasets can perform better than the full datasets.
>
> The author-reviewer discussion period will be closed in less than days, may I know if there are any other concerns? We truly value your insightful comment!
>
> Best,
> Authors from submission 243

---

> ### Author Response · Authors · 2023-11-22
> **Looking forward to your reply!**
>
> Dear reviewer G6x3:
>
> Thanks again for your efforts in evaluating our work. Considering the discussion period will conclude in less than hours, we would like to inquire whether your concerns are adequately addressed. Your insightful feedback plays a pivotal role in enhancing the clarity of our work!
>
> Best,
> Authors from submission 243

---

> > ### Comment · Reviewer_G6x3 · 2023-11-23
> > **Response to Rebuttal**
> >
> > Thanks for the detailed response, it addressed most of my concerns well. I keep my original acceptance score.

---

> > > ### Author Response · Authors · 2023-11-23
> > > **Response to Reviewer G6x3**
> > >
> > > Thanks for the valuable comments and feedback, which help to improve the quality of our work.

---

### Official Review · Reviewer_8eCM · 2023-10-31

**Soundness:** 3 good
**Presentation:** 4 excellent
**Contribution:** 4 excellent
**Rating:** 8
**Confidence:** 4

**Summary:**

The paper introduces a novel method of dataset distillation. It introduces the concept of difficulty-aligned trajectory matching, which controls the difficulty of the patterns generated on the synthetic data by matching different training phases of the expert model.

Overall, the paper makes a substantial contribution to the field of dataset distillation, introducing a novel method and providing valuable insights. Despite the need for clarification on certain points and a deeper exploration into the method’s performance across different scenarios, the paper stands out for its clarity, quality, and originality. With sufficient details provided for reproducibility, this work paves the way for further research and development in trajectory matching-based dataset distillation algorithms.

**Strengths:**

The paper provides a novel and intuitive insight into the effect of matching different training trajectories on the quality of the distilled dataset. The observation in the paper is insightful for the trajectory matching based dataset distillation. The paper is well-written and organized, with clear problem formulation, method description, and experimental setup.

Clarity, Quality, Novelty And Reproducibility:
The paper is clearly written, and the idea seems novel. The implementation details are provided for reproducing

**Weaknesses:**

Please refer to the Questions.

**Questions:**

I have a few questions and concerns about the method:

Q1. Manually setting lower bound and upper bound for the matching range is somehow incremental.

Q2. From Table 2a, the performance of “complex” networks is worse than 3-layer convolutional networks. While in table 2, the performance of “complex” networks is better than the 3-layer convolutional networks. It would be interesting to see the performance across architectures with the increasing number of IPC.

Q3. How is the synthetic dataset performance on the downstream tasks, such as object detection?

---

> ### Author Response · Authors · 2023-11-16
> **Response to Reviewer 8eCM (1/2)**
>
> > Q1. Manually setting lower bound and upper bound for the matching range is somehow incremental.
>
> We thank the reviewer for the constructive feedback. Although we still need to manually decide the matching range, we provide an efficient and insightful guidance for tuning it, which hasn't been introduced in previous works. The insights and details of our strategy are introduced as follows.
>
> **Insights**
> + By matching training trajectories, we expect the models trained on the distilled dataset can learn as many hard patterns as possible, without weakening their ability to identify easy (basic) patterns.
> + Considering DNNs tend to learn easier patterns in earlier training phases [1], we can tell if the model can learn easy patterns well from the synthetic dataset by observing its matching loss over early expert trajectories.
>
> **Findings**
> Through experiments, we find that:
> + Generating patterns that are too easy for the synthetic dataset will raise the matching loss over late trajectories. Because original hard patterns are ''overwritten'' by the generated easy patterns, resulting in information loss for the surrogate model to learn hard patterns through the synthetic data.
> + Accordingly, generating patterns that are too hard for the synthetic dataset will raise the matching loss over early trajectories.
>
> **Method**
> Based on the above observation, we can quickly find an appropriate matching range by observing the matching loss.
> 1. we first set the lower bound $T^-$ = 0 and upper bound $T^+$ =  10 to perform the distillation for 50 iterations, where the matching loss over a distillation-uninvolved expert trajectory is recorded.
> 2. Then we simultaneously increase the values of $T^-$ and $T^+$ by plus 10 until the distillation will not increase the matching loss over late expert trajectories (to avoid generating patterns that are too easy).
> 3. Subsequently, we increase the value of  $T^+$  by plus 10 until the distillation increases the matching loss over the early expert trajectories (to avoid generating patterns that are too hard).
>
> **Performance**
> Compared with grid search, our strategy costs much less resource since we only need to perform the distillation for 50 iterations and the synthetic steps $N$ can also be smaller here. For example, it **only takes around
> 5 minutes to find an appropriate matching range for CIFAR-10 IPC=50**.
>
> We have also tried to use grid search to decide the matching range, which *costs much more resources but only brings less than 0.6% improvement compared with our strategy*, showing the effectiveness and efficiency of our approach. The above contents are introduced in Section A.5 and Section A.6.
>
> **Conclusion**
> Our strategy **reduces the resource consumption of finding an appropriate matching range** and provides the first guidance for adjusting the matching range.
>
> [1] A closer look at memorization in deep networks, ICML 2017.
>
> ---
>
> > Q2. From Table 2a, the performance of “complex” networks is worse than 3-layer convolutional networks. While in Table 2, the performance of “complex” networks is better than the 3-layer convolutional networks. It would be interesting to see the performance across architectures with the increasing number of IPC.
>
> We compare the performance across architectures in Table 5, where the evaluations are performed with IPC=1, 50, 1000. Here we additionally report the evaluation on IPC=500 to see the trend more clearly. In the experiments reported below, we use ConvNet to perform the distillation, then the distilled datasets are evaluated with networks of various architectures.
>
> | IPC      | 10        | 50        | 500       | 1000      |
> | -------- | --------- | --------- | --------- | --------- |
> | ConvNet  | **68.28** | **76.08** | 83.5      | 85.50     |
> | ResNet18 | 48.66     | 66.27     | **84.18** | **87.22** |
> | VGG11    | 45.93     | 59.43     | 81.56     | 84.65     |
> | MLP      | 33.39     | 33.29     | 48.30     | 52.40     |
>
> According to the table, **ConvNet performs best in low IPC cases** such as 10 and 50. This is because, when IPC is low, we need to generate a large amount of easy patterns on the synthetic data, which will change the images drastically (Figure 4, 5). Thus the distilled datasets are more biased to the distillation backbone (ConvNet) in low IPC cases, making it perform better than 'complex' networks.
>
> However, **as IPC increased to 500 and 1000, complex networks (ResNet18) performs better**. Our method generates only hard patterns in high IPC cases, which make much smaller changes to the images (Figure 4, 5). Accordingly, the overfitting problem is alleviated; such that 'complex' networks now can perform better than the distillation backbone (ConvNet).
>
> Please let us know if we misunderstood the question. (It seems like there is a typo in the latter 'table 2' in the question?)

---

> ### Author Response · Authors · 2023-11-16
> **Response to Reviewer 8eCM (2/2)**
>
> > Q3. How is the synthetic dataset performance on the downstream tasks, such as object detection?
>
> Generally, the backbones used for object detection are pre-trained on large-scale datasets such as ImageNet with rather high resolution. Currently, we can not distill datasets with such a large scale due to the limitation of the compute resource. Therefore, we turn to evaluating the performance of transfer learning, which can reflect whether the representations learned by models can generalize well in downstream tasks.
>
> **Settings**
> 1. We first train models on:
> 	+  full dataset (CIFAR-10, IPC=5000).
> 	+  random selected data (CIFAR-10, IPC=500).
> 	+  data selected by Glister [2] (CIFAR-10, IPC=500).
> 	+  data selected by Forgetting [3] (CIFAR-10, IPC=500).
> 	+  our distilled dataset (CIFAR-10, IPC=500).
> 2. Then we froze all the parameters of the trained models except the ones in the fully connected layer.
> 3. After this, linear probing tasks are performed on STL-10 [1].
>
> **Results**
>
> |                  |Full Dataset|Random Selection | Glister |Forgetting |Ours|
> |:----------------:|:---------:|:---------:|:------------:|:------------:|:------------:|
> |       IPC        |5000 |   500    |      500      |500 |500|
> | Ratio |100%|   10%    |    10%    |10% |10%|
> | Acc on CIFAR-10 |84.8 |   73.2    |    72.5    | 74.6|83.5|
> | Acc on STL-10 (Transfer)    |72.8 |   62.41     |       60.83       |63.32 |**69.83** |
>
> **Conclusion**
> As the results reported above, the model pre-trained on our distilled dataset generalizes well: it performs similarly to the models pre-trained with full CIFAR-10 datasets. This reflects the models trained with our distilled datasets are capable of downstream tasks.
>
> **In addition, we will release all the distilled datasets to allow the community to explore their applications in downstream tasks.**
>
> [1] An analysis of single-layer networks in unsupervised feature learning, AISTATS 2011.
> [2] Glister: Generalization based data subset selection for efficient and robust learning, AAAI 2021.
> [3] An empirical study of example forgetting during deep neural network learning, ICLR 2018.

---

> ### Comment · Reviewer_8eCM · 2023-11-20
> **Raise my score to 8!**
>
> Your responses addressed most of my concerns. I will raise my score to 8. Again, realizing lossless dataset distillation is an important task, and keep digging it!

---

> > ### Author Response · Authors · 2023-11-20
> > **Response to Reviewer 8eCM**
> >
> > Thanks for the acknowledgment and valuable feedback. Lossless dataset distillation indeed is a meaningful topic. We will further explore it, trying to reduce the distillation cost and improve the generalizability.
> >
> > By the way, maybe there is an OpenReview system issue or delay, we find the score is still 6. Hopefully and we would appreciate it if the reviewer could have a look at it. Once again, thanks for your time and efforts on this work.

---

> > > ### Comment · Reviewer_8eCM · 2023-11-21
> > > **Confirmation: Raised my score to 8.**
> > >
> > > I have confirmed that I raised my score to 8. If you still cannot see it, please reach out to the AC.

---

> ### Author Response · Authors · 2023-11-21
> **Response to Reviewer 8eCM**
>
> The score is now updated. Thanks again for the efforts and time in this work.

---

### Official Review · Reviewer_5Mev · 2023-11-01

**Soundness:** 3 good
**Presentation:** 3 good
**Contribution:** 2 fair
**Rating:** 6
**Confidence:** 3

**Summary:**

As a compression algorithm, it remains unclear whether dataset distillation can be used to fully recover the full training accuracy of original dataset. The authors in this paper propose to address this issue through specifically aligning the difficult trajectories using part of the data. The experiment result show promising results.

**Strengths:**

+ The proposed framework is quite well motivated, and the experiment results aligns well with the observation
+ In the experiments, the authors are the first to demonstrate that distilled synthetic data can outperform real training dataset
+ The authors perform extensive studies and analysis on the algorithm

**Weaknesses:**

- The algorithm is extending MTT, although important, but there is not much algorithmic novelty
- In table 4, authors only compared with some simple baselines for cross-architecture generalization. It would be great if the authors can have a more thorough comparison with other dataset distillation algorithms as well.

**Questions:**

See above.

---

> ### Author Response · Authors · 2023-11-16
> **Response to Reviewer 5Mev**
>
> > Q1. The algorithm is extending MTT, although important, but there is not much algorithmic novelty
>
> We thank the reviewer for acknowledging the importance of our approach to the community. For clarity, we reiterate our novel contributions below:
>
> **Insight 1**
> We find matching earlier trajectories yields easier patterns. Moreover, we empirically show that easy patterns should not be used in high-IPC settings.
> **Algorithm innovation 1**
> Based on the above findings, we set a **lower bound on the matching range**, allowing TM-based methods to be effective in high IPC cases and achieving lossless performance for the first time.
>
> **Insight 2**
> We find generating only easy patterns in the early distillation phase makes it easier for the surrogate model to match the expert trajectories, helping to improve the stability of the distillation.
> **Algorithm innovation 2**
> Unlike previous TM-based methods that use a fixed upper limit to constrain the matching range, we propose to use **a floating upper bound** which is gradually increased as the distillation progresses. This makes the distillation stable enough such that we can **optimize soft labels** during the distillation, bringing non-trivial improvement across all IPC cases.
>
> **Insight 3**
> We find adding easy patterns brings more improvement when IPC is small. Accordingly, using easy samples to initialize the synthetic dataset can speed up the convergence of optimization when the size of the synthetic dataset is small (Section 4.3).
> **Algorithm innovation 3**
> Unlike previous TM-based methods that initialize synthetic dataset by random selection, we **use easy samples filtered by an expert model to initialize the synthetic data**. Our initialization can significantly speed up the distillation; it shortens the time required for optimization convergence to 5/8 of the original (Figure 3 (a)).
>
> Our innovations to the algorithm are novel, effective and insightful, which can help the community have a better understanding of the mechanism of trajectory-matching based distillation methods.
>
> ---
>
> >Q2.  Comparison of cross-architecture generalization with other dataset distillation methods.
>
> We appreciate the reviewer for caring about the generalizability; this is an important metric for evaluating a distillation algorithm.
>
> In Table 4, we compare the generalizability in high IPC settings. Since previous distillation methods perform similar or even worse than random selection when IPC is high (Figure 1 (d), [1, 2]), we choose to compare DATM with core-set selection methods, which are stronger baselines in high IPC cases. The results show our distilled datasets generalize well across various architectures in the regime of high IPC.
>
> In Table 2 (a), we compare the generalizability with two representative dataset distillation methods in low IPC settings (CIFAR-100, IPC=50). For convenience, the results are copied and reported below.
>
> |          Method        |ConvNet|ResNet 18 | VGG | AlexNet |
> |:----------------:|:---------:|:---------:|:------------:|:------------:|
> |       Random        |33.46 |   31.95    |      32.18      |26.65      |
> | MTT |45.68|   42.56    |    41.22    |40.29      |
> | FTD |48.90 |   46.65    |    43.24    |42.20      |
> | **Ours**    |**55.03** |   **51.71**     |       **45.38**       |**45.74**      |
>
>  Conclusion:
>  + **Our method performs best on four networks with different architectures, reflecting the outstanding generalizability of our distilled dataset.**
>
> [1] DC-BENCH: Dataset Condensation Benchmark, NeurIPS 2022.
> [2] Dataset Quantization, ICCV 2023.

---

> ### Author Response · Authors · 2023-11-20
> **Further Discussions with Reviewer 5Mev**
>
> Dear reviewer 5Mev:
>
> Thanks for your time and efforts in reviewing our paper. We detailed our algorithm innovation and provided a comprehensive study on generalizability. Hope our rebuttal has addressed your concerns.
> As the discussion period is nearing its end, please feel free to let us know if you have any other concerns. Thanks!
>
> Best,
> Authors from submission 243

---

> ### Author Response · Authors · 2023-11-21
> **Summary of Response**
>
> Dear Reviewer 5Mev,
>
> Considering the limited time available, and in order to save the reviewer's time, we summarize our responses here.
>
>  1. **\[Algorithm Innovation\]**
> 	**Question**: The algorithm is extending MTT, although important, but there is not much algorithmic novelty
>
> 	**Response**: We propose to **initialize synthetic data with easy samples**, **constrain the matching range** according to the IPC, **generate only easy patterns in the early distillation phase**, and **optimize soft labels** during the distillation. Our algorithm innovations are novel and insightful.
>
>  2. **[ Generalization Evaluation]**
> 	**Question**: It would be helpful to compare the generalization with other dataset distillation methods.
>
> 	**Response**: **Our method has better generalizability compared with previous distillation methods**. We report a comparison with previous distillation methods in Table 2 (a). Results show our method performs the best across four networks with different architectures, outperforming FTD by 4.22% on average.
>
> The author-reviewer discussion period will be closed in less than days, may I know if there are any other concerns? We truly value your insightful comment!
>
> Best,
> Authors from submission 243

---

> ### Author Response · Authors · 2023-11-22
> **Looking forward to your reply!**
>
> Dear reviewer 5Mev:
>
> Thanks again for your efforts in reviewing this work. As the discussion period will be closed in less than hours, may we know if the concerns are fully addressed? Your insightful comments and feedback are really important for us to improve the quality of our work!
>
> Best,
> Authors from submission 243

---

### Meta-Review · Area_Chair_6UC1 · 2023-12-08

**Metareview:**

This paper presents an interesting approach towards lossless dataset distillation. Based on observations that the effective size of synthetic datasets varies vastly depending on different stages of the training trajectory, the authors proposed to employ different distillation datasets for distillation from early and late training trajectories, and achieved significant improvements over existing baselines even under high IPCs.

All reviewers were positive about the paper. They appreciated the analysis presented in the paper and rebuttal, and agreed that the proposed approach is reasonable and the results are convincing. AC agrees with the reviewers that the paper provides meaningful contributions and insights to the field and hence recommends acceptance. The authors should include clarifications and additional results presented during the rebuttal to the camera-ready version of the paper.

**Justification For Why Not Higher Score:**

The paper presents significant empirical observations justified by convincing experiments. Yet, the analysis is limited to CNN-based backbones, which makes the generalizability of the method a bit uncertain.

**Justification For Why Not Lower Score:**

N/A

---

### Decision · Program_Chairs · 2024-01-16

Accept (poster)